# ⤳Attention Sinks: A 'Catch, Tag, Release' Mechanism for Embeddings

**Stephen Zhang, Mustafa Khan, Vardan Papyan**
University of Toronto, Vector Institute
{stephenn.zhang, mr.khan}@mail.utoronto.ca, vardan.papyan@utoronto.ca

## Abstract

Large language models (LLMs) often concentrate their attention on a few specific tokens referred to as *attention sinks*. Common examples include the first token, a prompt-independent sink, and punctuation tokens, which are prompt-dependent. While the tokens causing the sinks often lack direct semantic meaning, the presence of the sinks is critical for model performance, particularly under model compression and KV-caching. Despite their ubiquity, the function, semantic role, and origin of attention sinks—especially those beyond the first token—remain poorly understood. In this work, we conduct a comprehensive investigation demonstrating that attention sinks: *catch* a sequence of tokens, *tag* them using a common direction in embedding space, and *release* them back into the residual stream, where tokens are later retrieved based on the tags they have acquired. Probing experiments reveal these tags carry semantically meaningful information, such as the truth of a statement. These findings extend to reasoning models, where the mechanism spans more heads and explains greater variance in embeddings, or recent models with query-key normalization, where sinks remain just as prevalent. To encourage future theoretical analysis, we introduce a minimal problem which can be solved through the 'catch, tag, release' mechanism, and where it emerges through training.

## 1 Introduction

### 1.1 'Catch, Tag, Release' in Aquatic Conservation

In marine biology, the *'catch, tag, release'* mechanism is a vital tool for tracking fish populations. A fish is caught, fitted with a tracking tag encoding critical information, and then released back into the water stream, where it can be monitored over time. This process enables researchers to understand migration patterns and ecosystem interactions. Remarkably, LLMs exhibit a strikingly similar mechanism when processing tokens. To understand why, we first examine a recent discovery in LLM research.

### 1.2 The Prevalence of Attention Sinks

In many models, tokens often focus disproportionately on a select few positions in the sequence. Xiao et al. [2024] identified the first token as a common focal point, referring to it as an *attention sink*. Follow-up studies have shown that, beyond this fixed sink, additional ones can emerge based on the input – often appearing on punctuation [Yu et al., 2024, Sun et al., 2024, Cancedda, 2024].

Preserving these sinks has proven critical for retaining model performance in several key areas, including *KV-caching* [Xiao et al., 2024, Liu et al., 2024, Guo et al., 2024b, Willette et al., 2024], *quantization* [Lin et al., 2024, Son et al., 2024], and *pruning* [Zhang and Papyan, 2025a]. Motivated by these findings, researchers have begun to investigate the following question.

39th Conference on Neural Information Processing Systems (NeurIPS 2025).

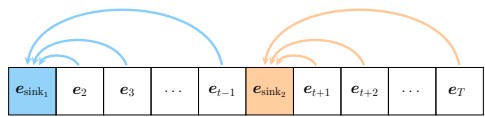

(a) **Catch**: Each box corresponds to a different token at the *input of the attention layer*, whose activation is denoted by $e_i$, and the arrows represent attention interactions. The *attention sinks* $e_{\text{sink}_1}$ and $e_{\text{sink}_2}$ *catch* the attention of tokens $e_2, e_3, \ldots, e_{t-1}$ and $e_{t+1}, e_{t+2}, \ldots, e_T$, respectively. This causes vertical bands to emerge in the attention weights $A$, as shown in Figure 2a.

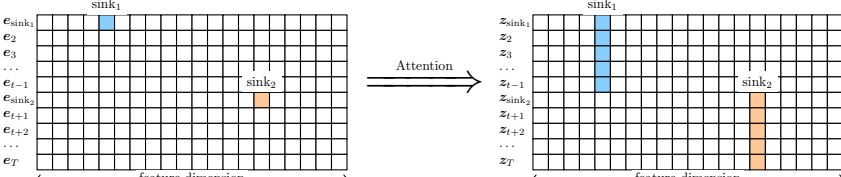

(b) **Tag**: The left grid shows the *attention value matrix* $V = [e_1; \ldots; e_T]$, where activation vectors $e_i$ are stacked vertically. The right grid shows the output of the attention layer $Z = [z_{\text{sink}_1}; z_2; \ldots; z_T] = AV$, with output vectors $z_i$ also stacked vertically. The value vectors of the sinks, $e_{\text{sink}1}$ and $e_{\text{sink}_2}$, are copied to all tokens that attend to them, thereby tagging them. These tags cause the token representation to cluster based on the sink they attended to, as revealed in the PCA plot in Figure 2c. The inputs to the attention layer, prior to the tagging, show no such clustering, as shown in Figure 2b.

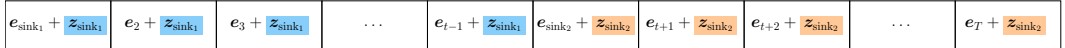

(c) **Release**: Each box corresponds to a different token at the *output of the attention layer*. The attention outputs are added to the residual stream as $e_i + z_{\text{sink}}$, creating common directions in representation space, in the form of the tags $z_{\text{sink}_1}$ and $z_{\text{sink}_2}$ shared across multiple tokens. These tags cause the token representations to cluster in deeper layers, as revealed in the PCA plot in Figure 2d.

Figure 1: An illustration of the 'catch, tag, release' mechanism.

## 1.3 Why Do Attention Sinks Emerge?

Existing answers generally fall into three main categories:

**To Create Implicit Attention Biases:** Attention layers lack explicit bias parameters. Attention sinks emerge as compensatory mechanisms that artificially introduce such biases, and they can be mitigated by incorporating explicit key or value bias parameters [Sun et al., 2024, Darcet et al., 2024, Gu et al., 2024].

**To Turn Off Attention Heads:** Attention heads are not needed for certain sequences. Attention sinks emerge as a learned, data-dependent mechanism that effectively disables them by capturing nearly all the attention [Bondarenko et al., 2023, Guo et al., 2024a].

**To Prevent Over-Mixing of Tokens:** Transformers are prone to excessive token mixing and rank collapse [Geshkovski et al., 2024, Barbero et al., 2024, 2025b]. First-token attention sinks help mitigate these issues by anchoring the sequence and limiting uncontrolled interaction across positions.

## 1.4 Open Questions

While each of the three perspectives shine a light on the role of *first-token* attention sink, they still leave the following questions unanswered:

Q1: *Why do attention sinks emerge in later tokens?*

None of the perspectives addresses this question. These sinks are sequence-dependent [Yu et al., 2024, Cancedda, 2024], contradicting the bias perspective, and the presence of multiple sinks seems unnecessary from both the active-dormant and over-mixing perspective.

Q2: *Do the representations of attention sinks—despite corresponding to semantically meaningless tokens like punctuation—nonetheless encode meaningful information?*

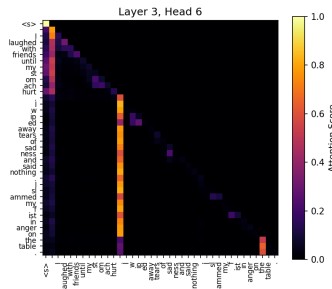

(a) **Attention Weights.** Two attention sinks catch the attention of subsequent tokens in the sequence.

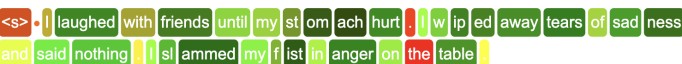

(b) **PCA on input to the attention layer.** Tokens exhibit no clustering.

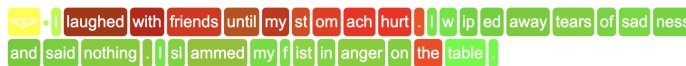

(c) **PCA on the output of the attention layer.** Tokens cluster according to their *attended sink*: those attending to the first sink are shaded red, while those attending to the second sink are shaded green.

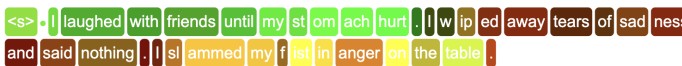

(d) **PCA on the residual stream in a deeper layer.** Tagged tokens propagate through the residual stream, clustering in a deeper layer based on their previously attended sink. Tokens that attended to the first sink are green, while those attending to the second sink are brown and yellow.

Figure 2: **Qualitative analysis of the 'catch, tag, release' mechanism.** The second and third subplots use PCA-based coloring of embeddings, described in Section 2. Appendix A presents additional measurements across a wide range of models, layers, attention heads, and prompts, including chain-of-thought [Wei et al., 2022], and zero-shot chain-of-thought [Brown et al., 2020].

This question remains largely unexplored [Yu et al., 2024], leading to conflicting lines of research, with some studies exploring how sinks can be preserved [Xiao et al., 2024, Son et al., 2024], while others investigating how they can be dispersed or removed [Sun et al., 2024, Yu et al., 2024, Zuhri et al., 2025, Fu et al., 2025, Kang et al., 2025].

> Q3: *Are there differences in the number of attention sinks between pretrained LLMs and those fine-tuned for reasoning tasks [DeepSeek-AI, 2025]?*

If attention sinks play a functional role in reasoning, one would expect them to be more prevalent in models optimized for such purposes. Furthermore, the locations of these sinks may align with semantically meaningful boundaries, segmenting content in ways that support structured reasoning.

> Q4: *How does query-key normalization affect attention sinks?*

It is conceivable that query-key (QK) normalization [Henry et al., 2020] alters the effective temperature of the softmax distribution in attention layers. This, in turn, may smoothen or sharpen attention weights, potentially affecting the formation of attention sinks.

## 1.5 Contributions

Through an empirical study, we answer the questions posed above and establish the following claims:

> A1: Attention sinks implement a *'catch, tag, release'* mechanism, the steps of which are detailed in Figure 1 (Sections 2, 3).

> A2: Probing experiments reveal that the resulting tags are not arbitrary – they encode semantically meaningful information, such as the truth value of a statement (Section 4).

> A3: Compared to *pretrained* models, *DeepSeek-distilled* models exhibit the mechanism across more attention heads and explain a greater proportion of variance in the embeddings, indicating a stronger and more pronounced instantiation of the mechanism (Section 5).

> A4: Attention sinks remain prevalent in models with QK normalization, despite the normalization explicitly imposed on tokens prior to computing attention scores. (Section 6).

To support future theoretical analysis, we introduce a minimal problem that is solvable via the explicit use of the 'catch, tag, release' mechanism and demonstrate empirically that the mechanism naturally emerges through standard training.

## 2 Visualizing the 'Catch, Tag, Release' Mechanism

This section presents a visual exploration of the 'catch, tag, release' mechanism. Figure 2 illustrates a representative example selected for clarity, but similar behavior consistently emerges across a wide range of models, prompts, layers, and attention heads. To support this generality, we include additional visualizations in Appendix A.

**Evidence for Catch** As illustrated in Figure 1a, the evidence of the *catch* mechanism amounts to showing the existence of an attention sink. We therefore feed a prompt to the PHI-3 MEDIUM model [Abdin et al., 2024a], and save the attention weights of layer 3, head 6 and plot them in Figure 2a. The visualization shows that there are two sinks that are catching the attention of subsequent tokens.

**Evidence for Tag** As shown in Figure 1b, demonstrating the *tagging* mechanism requires verifying token clustering based on the attended sink. Following Oquab et al. [2024], we compute the top two or three (depending on the setting) principal components of the $d \times d$ covariance matrix of the attention head outputs, where $d$ is the embedding dimension, and project the activations onto this basis. The projected values are then normalized to $[0, 255]$ and mapped to RG or RGB channels. Figure 2c illustrates the results, showing a clear grouping of tokens by their attention sinks at the *output of the attention head*. In contrast, Figure 2b depicts the absence of such grouping at the *input to the attention head*.

**Evidence for Release** As depicted in Figure 1c, evidence for *release* amounts to showing that tokens in the residual stream in deeper layers have the same clustering behavior as exhibited in an earlier attention head output. We therefore hook the inputs into the feedforward network in layer 17, and apply the same PCA projection and normalization step as described earlier. The results, presented in Figure 2d, reveal a similar grouping, providing evidence that the model has utilized the tags generated in the earlier layers to cluster the embeddings.

## 3 Measuring the 'Catch, Tag, Release' Mechanism

We provide a quantitative analysis to substantiate the presence of the 'catch, tag, release' mechanism across a wide range of model families, including QWEN 2.5 [Yang et al., 2024], PHI-3 [Abdin et al., 2024a], LLAMA-3 [Grattafiori et al., 2024], and MISTRAL 7B [Jiang et al., 2023]. The analysis is aggregated over 170 prompts collected by Gu et al. [2024], truncated to 150 tokens.

### 3.1 Identifying Attention Sinks and Their Associated Tags

We leverage the metric proposed by Gu et al. [2024] to identify which tokens are attention sinks. Letting $\boldsymbol{A}$ denote the attention weights (i.e. softmax probabilities) of an attention head, token $t$ is identified as an attention sink for that head if and only if:

$$\alpha_t := \frac{1}{T - t + 1} \sum_{k=1}^{T} \boldsymbol{A}_{k,t} > \epsilon, \tag{1}$$

where $T$ is the length of the prompt and $\epsilon$ is a predetermined threshold that is set to $\epsilon = 0.2$ for all the following experiments[1].

For any token $t$ designated as an attention sink, its tag is defined to be its value vector extracted from the attention head's value matrix: $\boldsymbol{v}_t = \boldsymbol{V}_{t,:} \in \mathbb{R}^{d_{\text{head}}}$, where $d_{\text{head}}$ is the head dimension.

### 3.2 Quantifying Variance Explained by Tags

Suppose $n$ tokens are identified as attention sinks. We concatenate their tags into a matrix $\mathbf{V}_{\text{tag}} \in \mathbb{R}^{n \times d_{\text{head}}}$ and apply Principal Component Analysis (PCA):

$$\mathbf{V}_{\text{tag}}^{\top} \mathbf{V}_{\text{tag}} = \mathbf{U} \mathbf{D} \mathbf{U}^{\top},$$

---

[1] A sensitivity analysis of the threshold choice is provided in Appendix F.

where $U$ is the matrix of eigenvectors and $D$ is the diagonal matrix of eigenvalues. The matrix $UU^\top$, which has rank $n$, defines a projection onto the subspace spanned by the tags. We take the output of the attention head, given by $AV$ (where $A$ and $V$ denote the attention weights and values, respectively), and project it onto this subspace, yielding the:

$$\text{Variance Explained} = \frac{\|AVUU^\top\|_F}{\|AV\|_F}. \tag{2}$$

This quantity is upper bounded by one with equality only if the attention head's output lies entirely within the tag subspace, providing a soft measure of how much of the output is explained by the tags.

Figure 3 below shows that 1-2 tags typically explain $20\% - 40\%$ of the variance in the outputs of the attention heads, but in many cases up to $70\%$. This provides evidence that the tokens are being tagged and that these tags contribute significantly to the tokens' representations.

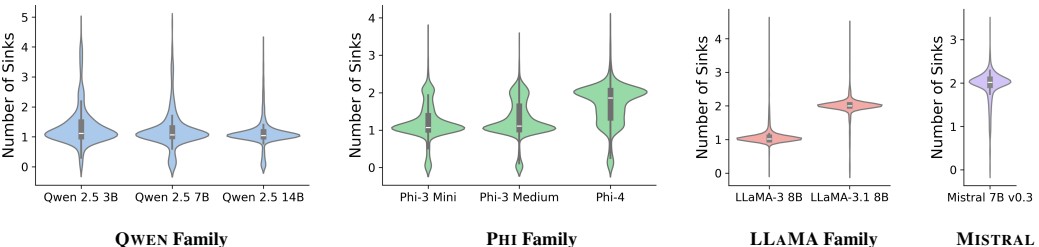

(a) **Attention Sink Counts.** Counts are computed for each head using Equation 1, and their distribution is visualized for each model using violin plots.

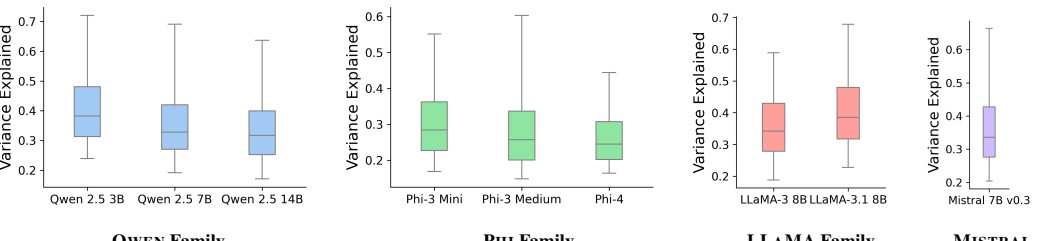

(b) **Variance Explained by Tags.** The metric is computed for each head using Equation 2, and summary statistics are shown for each model using box plots. Boxes indicate the 25th, 50th, and 75th percentiles, while whiskers represent the 5th and 95th percentiles.

Figure 3: Quantitative analysis of the 'Catch, Tag, Release' mechanism.

## 4 Investigating the Semantic Role of Tags

The previous section demonstrated that attention sinks tag tokens. This observation prompts a deeper question: do the distributed tags carry semantically meaningful information that is being *released* back into the residual stream as part of the release mechanism? To answer this question, we turn to a dataset designed to test semantic encoding.

### 4.1 Cities Dataset [Marks and Tegmark, 2024]

The dataset consists of prompts involving (`[CITY]`, `[COUNTRY]`) pairs that render a statement either true or false (see Figure 4). The goal is to assess whether the activations of individual tokens can be used to classify the True/False label of the statement. We focus specifically on the final period token – located at position $t$, indicated by the arrow in Figure 4 – as punctuation tokens frequently cause attention sinks.

> The city of Tokyo is in Japan.
> This statement is: TRUE.
> The city of Hanoi is in Poland.
> This statement is: FALSE.
> The city of `[CITY]` is in `[COUNTRY]`.
> This statement is:

Figure 4: `cities` prompt example.

## 4.2 Decomposing Activations into Tag and Non-Tag Components

The decomposition of the activation is given by:

$$z = \sum_{k=1}^{T} A_{t,k} V_{k,\,:} = z_{\text{tag}} + z_{\text{no tag}}$$

where:

$$z_{\text{tag}} = \sum_{k=1}^{T} \mathbf{1}_{[\alpha_k > \epsilon]} \cdot A_{t,k} V_{k,\,:} \qquad \text{and} \qquad z_{\text{no tag}} = \sum_{k=1}^{T} \mathbf{1}_{[\alpha_k < \epsilon]} \cdot A_{t,k} V_{k,\,:}\,,$$

and $\alpha_k$ was defined in Equation 1. This decomposition explicitly expresses the token's activations as the linear combination of tag and non-tag components.

## 4.3 Constructing Mass-Mean Probes Using Tags

First, we input $N_+$ true prompts and $N_-$ false prompts from the `cities` dataset to compute the class means:

$$\mu_{\text{tag}}^{+} = \frac{1}{N_+} \sum_{i \in \text{True}} z_{t,\text{tag}}^{(i)}, \qquad\qquad \mu_{\text{tag}}^{-} = \frac{1}{N_-} \sum_{i \in \text{False}} z_{t,\text{tag}}^{(i)},$$

$$\mu_{\text{no tag}}^{+} = \frac{1}{N_+} \sum_{i \in \text{True}} z_{t,\text{no tag}}^{(i)}, \qquad\qquad \mu_{\text{no tag}}^{-} = \frac{1}{N_-} \sum_{i \in \text{False}} z_{t,\text{no tag}}^{(i)},$$

Next, we compute similarly the within-class covariances, $\Sigma_{\text{tag}}, \Sigma_{\text{no tag}}$. These are used to generate two sets of mass-mean probes:

$$\theta_{\text{tag}}(z) = \sigma\left(z^\top \Sigma_{\text{tag}}^{-1}(\mu_{\text{tag}}^{+} - \mu_{\text{tag}}^{-})\right), \qquad\qquad \theta_{\text{no tag}}(z) = \sigma\left(z^\top \Sigma_{\text{no tag}}^{-1}(\mu_{\text{no tag}}^{+} - \mu_{\text{no tag}}^{-})\right),$$

where $\sigma(\cdot)$ is the logistic function. This closely follows the experimental setup of [Marks and Tegmark, 2024], with the key distinction that we decompose activations into tag and non-tag components, rather than using the full activation directly.

We employ a total of 600 prompts from the dataset: 400 of which are utilized to generate the probes and 200 for validation, ensuring each set contains an equal number of True and False statements. For each model, specific attention heads are identified where the $\theta_{\text{tag}}$ convey True/False information. Further details, including a taxonomy of which tokens were identified as tags for the probes are provided in Appendix J.

## 4.4 Comparing Probe Performance

Table 1 summarizes the performance of probes derived from the tag component of the activations $\theta_{\text{tag}}$, the non-tag component $\theta_{\text{no tag}}$, and the full activation $\theta_{\text{activation}}$. The superior performance of $\theta_{\text{tag}}$ confirms that the tags contain semantically meaningful information, while the disparity between $\theta_{\text{tag}}$ and $\theta_{\text{no tag}}$ demonstrates that the tags distribute information not present in the tokens. Notably, $\theta_{\text{tag}}$ can outperform the full-activation probes, suggesting that the tags can provide a *denoised* representation of the True/False direction.

| Probe | QWEN 2.5 | | | LLAMA-3 | LLAMA-3.1 |
|---|---|---|---|---|---|
| | 3B | 7B | 14B | 8B | 8B |
| $\theta_{\text{tag}}$ | 98% | 94.5% | 99.5% | 99.0% | 92.5% |
| $\theta_{\text{no tag}}$ | 50.0% | 50.0% | 50.5% | 56.5% | 50.0% |
| $\theta_{\text{activation}}$ | 50.0% | 83.0% | 60% | 97.0% | 86.0% |

Table 1: **Classification Accuracy of Probes.** The probe $\theta_{\text{tag}}$ is computed from the *tag* component of the activation, $\theta_{\text{no tag}}$ from the *non-tag* component, and $\theta_{\text{activation}}$ from the *full* activation.

# 5 Comparing Pretrained and Reasoning Models

DeepSeek-AI [2025] distilled the DEEPSEEK-R1 model into two widely used pretrained architectures: LLAMA 3.1 8B and QWEN 2.5 14B. In this section, we compare these *reasoning-distilled* variants to their original *pretrained* counterparts to examine how distillation for reasoning affects the emergence of the catch, tag, release mechanism.

| Pretrained Model | Reasoning-Distilled Variant |
|---|---|
| LLAMA 3.1 8B | DEEPSEEK-R1 LLAMA 8B |
| QWEN 2.5 14B | DEEPSEEK-R1 QWEN 14B |

Table 2: Summary of models investigated.

Figure 5 presents the average number of attention sinks, alongside heat maps of the variance explained by tags across attention heads and layers. Reasoning-distilled models exhibit more sinks, particularly in the case of QWEN, and also feature more attention heads with high variance explained by the tags. This suggests that the 'catch, tag, release' mechanism is more prominent in reasoning-distilled models than in their pretrained counterparts.

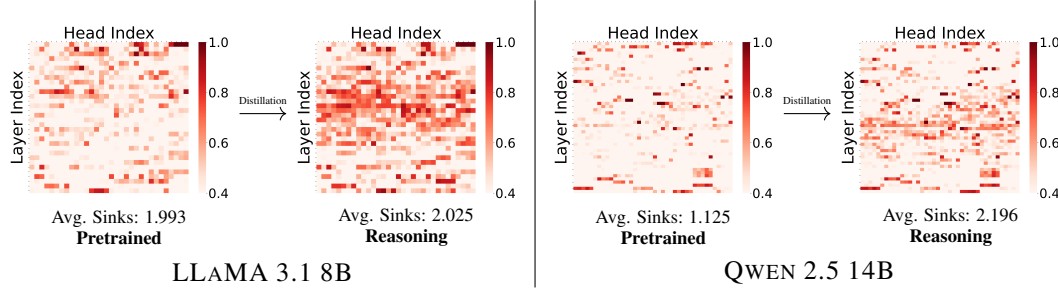

Figure 5: **'Catch, Tag, Release' in Pretrained vs. Reasoning Models.** The figure compares the number of attention sinks and the variance explained by tags in reasoning-distilled models and their pretrained counterparts, respectively.

# 6 Analyzing the Impact of Query-Key Normalization on Attention Sinks

Massive outlier activations refer to a phenomenon in which certain tokens in LLMs exhibit unusually large activation entries [Kovaleva et al., 2021, Dettmers et al., 2022, Puccetti et al., 2022, Hämmerl et al., 2023, Rudman et al., 2023, Crabbé et al., 2024]. These tokens tend to dominate attention computations, as their large values lead to high inner products with other tokens. This, in turn, draws a disproportionate share of attention to them, causing them to become *attention sinks* [Sun et al., 2024, Kaul et al., 2024, Guo et al., 2024a].

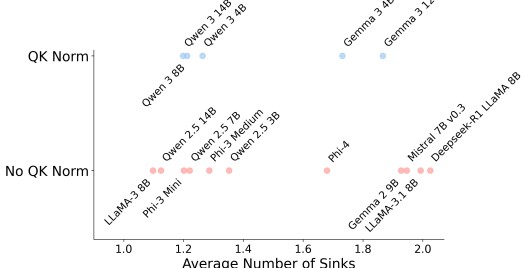

Figure 6: **Sink Count in Models with QK Norm.** For each model, the average number of sinks is computed across all attention heads and layers.

A recent architectural intervention that appears intended to counteract this is *QK normalization* [Henry et al., 2020], which normalizes query and key activations prior to the computation of attention scores. This normalization aims to reduce token magnitudes, thereby dampening the influence of activation outliers. One might therefore expect it to also reduce the number of attention sinks–and, as a result, suppress the 'catch, tag, release' mechanism we introduce in this paper.

Figure 6 reports the average number of sinks across all attention heads in models with and without query-key normalization. Surprisingly, the total number of sinks remains similar between the two settings, suggesting that QK normalization does not eliminate sink formation.

# 7 Establishing a Theoretical Foundation for 'Catch, Tag, Release'

In this section, we introduce a minimal problem that can be solved by explicitly leveraging the 'catch, tag, release' mechanism. We further show that this mechanism naturally emerges through optimization.

## 7.1 Setup

**Task:** Given a sequence of $T$ tokens, consisting of numbers, $x_i \in \mathbb{R}$, separated by a special `[SEP]` token:

$$\boldsymbol{x} = (x_1, ..., x_{t-1}, \texttt{[SEP]}, x_{t+1}, ..., x_T),$$

the objective is to compute the average of the numbers appearing after the `[SEP]` token. To increase the complexity of the task, the position of the `[SEP]` token, denoted by $t$, varies across different sequences.

**Embeddings:** The number tokens and `[SEP]` are embedded into:

$$\boldsymbol{e}_i = \texttt{Embed}(x_i) = \begin{bmatrix} x_i \\ -1 \end{bmatrix} \in \mathbb{R}^2, \quad i \neq t, \qquad \boldsymbol{e}_t = \texttt{Embed}(\texttt{[SEP]}) = \begin{bmatrix} s_{\texttt{num}} \\ -s_{\texttt{tag}} \end{bmatrix} \in \mathbb{R}^2,$$

respectively, where $s_{\texttt{num}}, s_{\texttt{tag}} \in \mathbb{R}$ are learnable parameters. The first coordinate of the embeddings represents the numbers, while the second coordinate represents the tag. The embeddings are concatenated into a matrix to form:

$$\boldsymbol{E} = \begin{bmatrix} \boldsymbol{e}_1 & \boldsymbol{e}_2 & \cdots & \boldsymbol{e}_T \end{bmatrix}^\top \in \mathbb{R}^{T \times 2}.$$

**Model:** The embeddings are passed as input to a two-layer transformer [Vaswani et al., 2017]:

$$\boldsymbol{H} = \text{Attention}(\boldsymbol{E}, \boldsymbol{E}, \boldsymbol{E}\boldsymbol{W}_V^1) + \boldsymbol{E} \tag{3}$$

$$f_\theta(\boldsymbol{x}) = \text{Attention}(\boldsymbol{h}_T^\top \boldsymbol{W}_Q^2, \boldsymbol{H}\boldsymbol{W}_K^2, \boldsymbol{H}\boldsymbol{W}_V^2), \tag{4}$$

parameterized by:

$$\theta = (\boldsymbol{W}_V^1, \boldsymbol{W}_Q^2, \boldsymbol{W}_K^2, \boldsymbol{W}_V^2, s_{\texttt{num}}, s_{\texttt{tag}}).$$

where the attention is causal and computes:

$$\text{Attention}(\boldsymbol{Q}, \boldsymbol{K}, \boldsymbol{V}) = \text{softmax}(\boldsymbol{Q}\boldsymbol{K}^\top)\boldsymbol{V}.$$

## 7.2 Main Result

Theorem 7.1 below captures how the 'catch, tag, release' mechanism explicitly solves the sequence averaging task, with the complete proof provided in Appendix E.

---

**Theorem 7.1 ('Catch, Tag, Release' Theory)** *Assume the learnable parameters, $\theta$, satisfy:*

$$s_{num} = 0, \quad \boldsymbol{W}_V^1 = \begin{bmatrix} 0 & 0 \\ 0 & -1 \end{bmatrix}, \quad \boldsymbol{W}_Q^2 \boldsymbol{W}_K^2 = \begin{bmatrix} 0 & b \\ 0 & d \end{bmatrix}, \quad \textit{where } d > 0, b \in \mathbb{R}, \quad \boldsymbol{W}_V^2 = \begin{bmatrix} 1 \\ 0 \end{bmatrix}.$$

*Then:*

$$f_\theta(\boldsymbol{x}) \xrightarrow{\;s_{tag} \to \infty\;} \frac{1}{T-t} \sum_{i=t+1}^{T} x_i$$

*for any $x_1, ..., x_T \in \mathbb{R}$. The model will have the following features for any sequence $\boldsymbol{x}$:*

- ***Catch:*** *The `[SEP]` forms an **attention sink**.*

- ***Tag:*** *The tokens after the `[SEP]` are tagged by the `[SEP]` token's values.*

- ***Release:*** *The tag is used to identify the tokens that should be averaged.*

---

### 7.3 Emergence of 'Catch, Tag, Release' Through Optimization

We train the model on $8,192$ sequences for $50$ epochs, using AdamW [Loshchilov and Hutter, 2019] with a learning rate of `5e-2` and weight decay of `1e-3`. Depicted in Figure 7 are the components of the 'catch, tag, release' mechanism that emerge with optimization.

**Catch:** Figure 7a depicts the attention weights of the first layer where the [SEP] forms an attention sink.

**Tag:** Figure 7b visualizes the attention head output, showing that the [SEP] token tagged all the subsequent tokens through their second coordinate, which exhibit a significantly larger magnitude.

**Release:** Figure 7c depicts the attention weights of the second layer, demonstrating how the model only averages over the tokens that were tagged, i.e., tokens following [SEP].

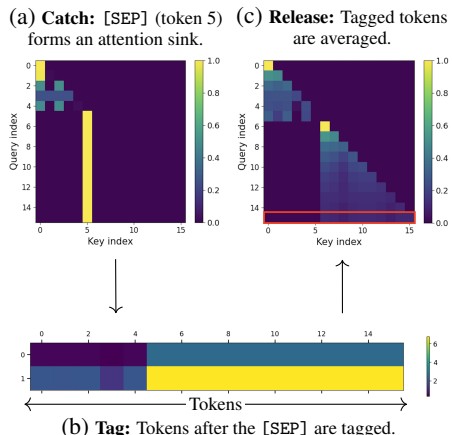

(a) **Catch:** [SEP] (token 5) forms an attention sink.

(c) **Release:** Tagged tokens are averaged.

(b) **Tag:** Tokens after the [SEP] are tagged.

Figure 7: **Emergence Through Optimization.** The emergence of the 'catch, tag, release' mechanism in the theoretical model through optimization. The [SEP] token occurs at index 5 for the prompt used to generate the plots.

## 8 Additional Related Works

**Understanding the Origins of Attention Sinks** A recent line of investigation considers the role of *positional encodings*. Guo et al. [2024b] showed that first-token sinks emerge even in the absence of positional encodings, suggesting that their formation is not solely a positional artifact. In contrast, other recent work has demonstrated that the varying frequencies in RoPE [Su et al., 2021] are exploited by models to produce distinct attention patterns, which may be either positional or semantic in nature [Barbero et al., 2025a].

Another hypothesis is that these phenomena may be *optimizer-induced*. Both Kaul et al. [2024] and Guo et al. [2024a] show that the Adam optimizer [Kingma and Ba, 2015] leads to attention sinks and outlier features. The former, along with many other works [Hu et al., 2024, Nrusimha et al., 2024, He et al., 2024], develop techniques to *prevent outlier dimensions* from forming, which has been linked to the formation of attention sinks.

**Connection to Rank Collapse** A closely related phenomenon is *rank collapse*, where token representations progressively lose dimensionality with depth Anagnostidis et al. [2022], Geshkovski et al. [2024], Barbero et al. [2024], Naderi et al. [2025], Kirsanov et al. [2025]. This collapse may itself stem from the same 'catch, tag, release' dynamics – attention sinks catch tokens, imprint shared tags, and release them into the residual stream, where the representations eventually collapse into the subspace defined by these tags.

## 9 Conclusion

This work uncovers and formalizes *'catch, tag, release'*, a ubiquitous mechanism in LLMs mediated by attention sinks, demonstrating that sinks are not mere quirks of attention maps, but instead implement a functional tagging system that propagates semantically meaningful information across tokens. The mechanism persists across diverse model families, intensifies in models fine-tuned for reasoning, and remains robust even under architectural modifications such as QK normalization. Beyond describing this behavior, we introduce a theoretical construction where the mechanism arises naturally and proves sufficient for solving a well-defined task. This offers a foundation for deeper mechanistic investigations into token interactions and the role of implicit memory in transformers.

## Acknowledgments and Disclosure of Funding

We would like to thank David Glukhov for his helpful feedback and discussion. We acknowledge the support of the Natural Sciences and Engineering Research Council of Canada (NSERC). This research was supported in part by the Province of Ontario, the Government of Canada through CI-FAR, and industry sponsors of the Vector Institute (`www.vectorinstitute.ai/partnerships/current-partners/`). This research was also enabled in part by support provided by Compute Ontario (`https://www.computeontario.ca`) and the Digital Research Alliance of Canada (`https://alliancecan.ca`).

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

# A  Additional Visualizations

In this section we provide further visualizations to the one presented in Section 2 across a variety of prompts and models.

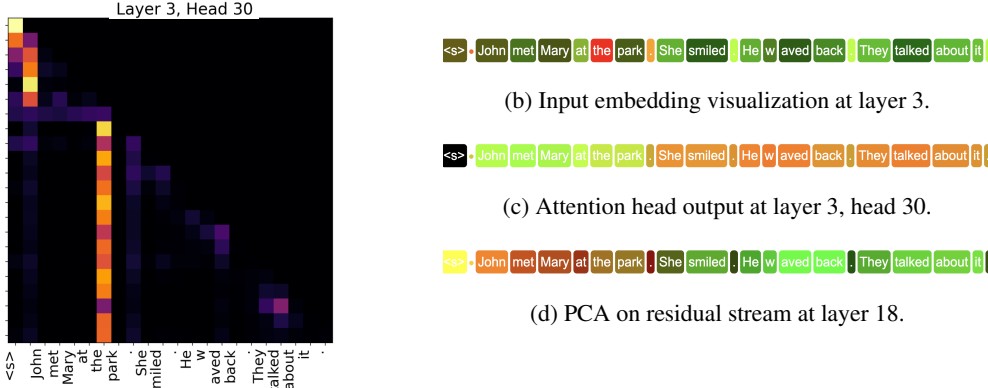

(a) Attention probabilities at layer 3, head 30.

(b) Input embedding visualization at layer 3.

(c) Attention head output at layer 3, head 30.

(d) PCA on residual stream at layer 18.

Figure 8: Visualization of the 'catch, tag, release' mechanism on a sample referential prompt on the PHI-3 MEDIUM model.

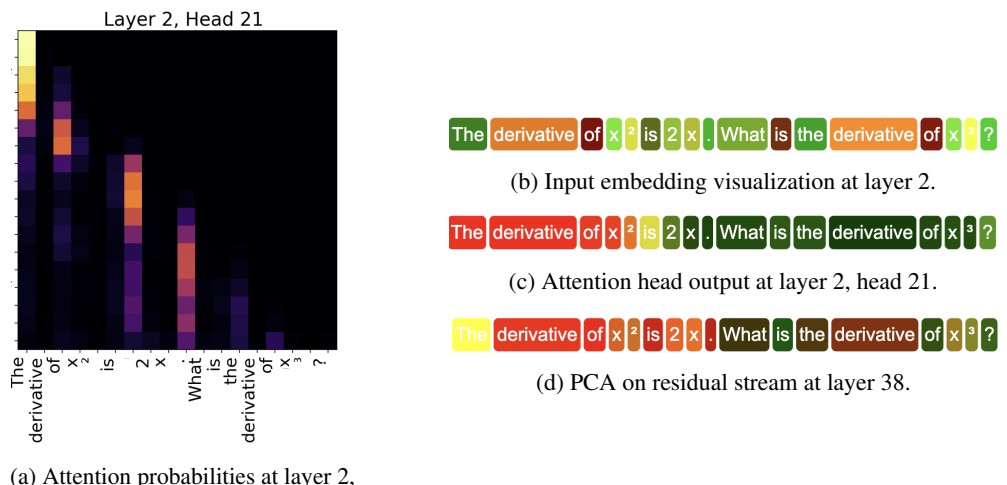

(a) Attention probabilities at layer 2, head 21.

(b) Input embedding visualization at layer 2.

(c) Attention head output at layer 2, head 21.

(d) PCA on residual stream at layer 38.

Figure 9: Visualization of the 'catch, tag, release' mechanism on a sample one-shot learning prompt on the QWEN 2.5-32B-INSTRUCT model.

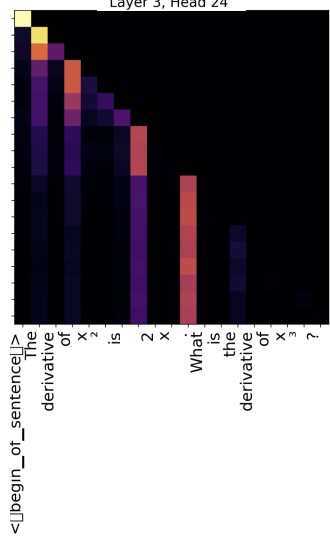

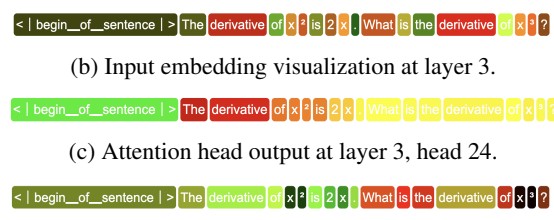

(b) Input embedding visualization at layer 3.

(c) Attention head output at layer 3, head 24.

(d) PCA on residual stream at layer 14.

(a) Attention probabilities at layer 3, head 24.

Figure 10: Visualization of the 'catch, tag, release' mechanism on a sample one-shot learning prompt on the DEEPSEEK-MATH-7B-INSTRUCT.

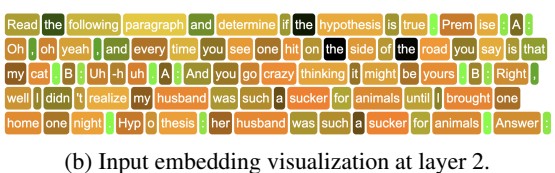

(b) Input embedding visualization at layer 2.

(c) Attention head output at layer 2, head 16.

(d) PCA on residual stream at layer 8.

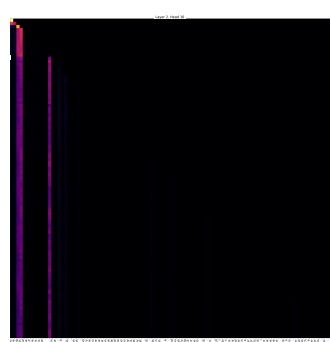

(a) Attention probabilities at layer 2, head 16.

Figure 11: Visualization of the 'catch, tag, release' mechanism on a longer prompt on the QWEN 2.5-14B-INSTRUCT model.

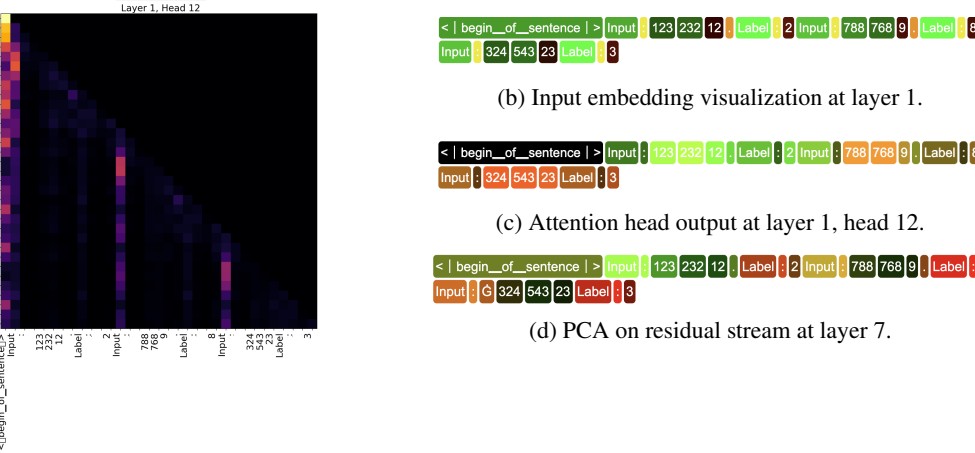

(a) Attention probabilities at layer 1, head 12.

(b) Input embedding visualization at layer 1.

(c) Attention head output at layer 1, head 12.

(d) PCA on residual stream at layer 7.

Figure 12: Visualization of the 'catch, tag, release' mechanism on a sequence averaging prompt on the DEEPSEEK-R1-DISTILL-LLAMA-8B model.

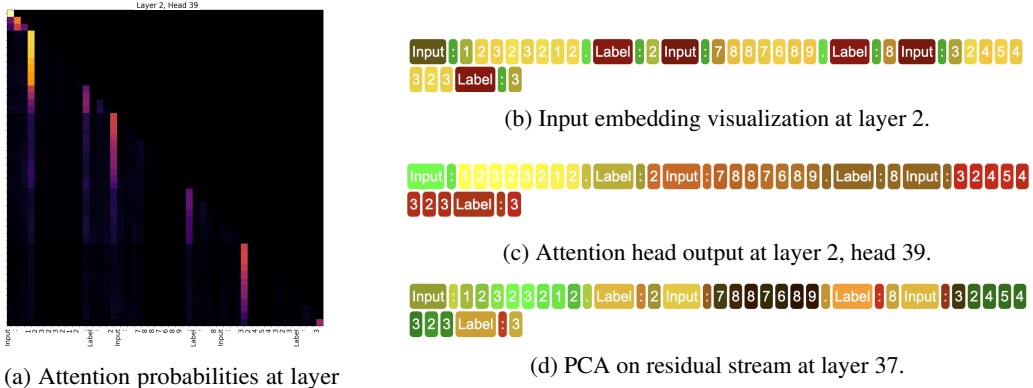

(a) Attention probabilities at layer 2, head 39.

(b) Input embedding visualization at layer 2.

(c) Attention head output at layer 2, head 39.

(d) PCA on residual stream at layer 37.

Figure 13: Visualization of the 'catch, tag, release' mechanism on a sequence averaging prompt on the QWEN2.5-14B-INSTRUCT model.

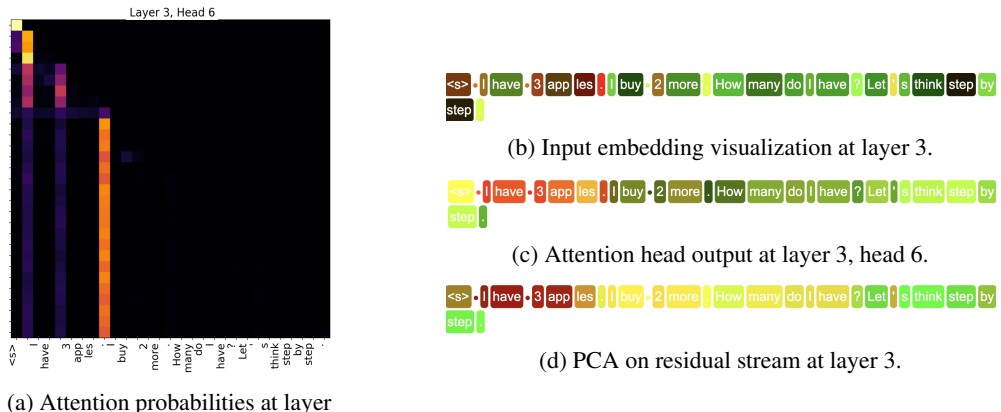

(a) Attention probabilities at layer 3, head 6.

(b) Input embedding visualization at layer 3.

(c) Attention head output at layer 3, head 6.

(d) PCA on residual stream at layer 3.

Figure 14: Visualization of the 'catch, tag, release' mechanism on a Zero-Shot Chain of Thought (CoT) prompt on the PHI-3 MEDIUM model.

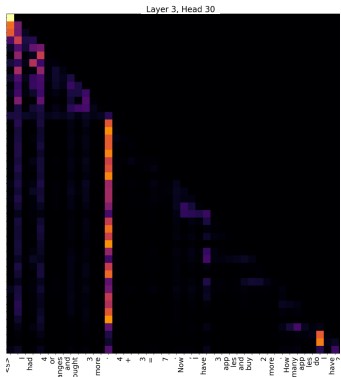

(a) Attention probabilities at layer 3, head 30.

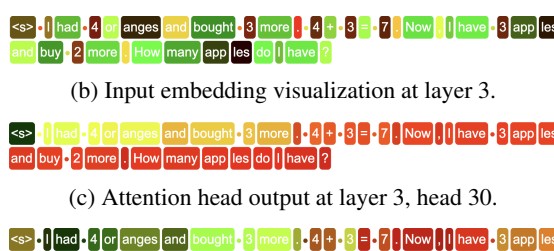

(b) Input embedding visualization at layer 3.

(c) Attention head output at layer 3, head 30.

(d) PCA on residual stream at layer 16.

Figure 15: Visualization of the 'catch, tag, release' mechanism on a one-shot math prompt on the PHI-3 MEDIUM model.

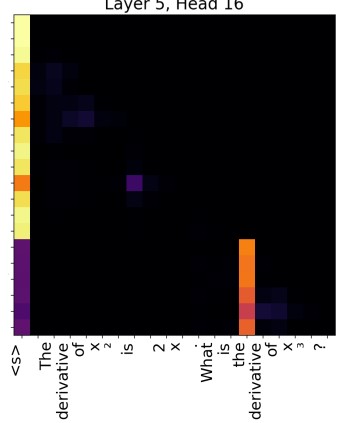

(a) Attention probabilities at layer 5, head 16.

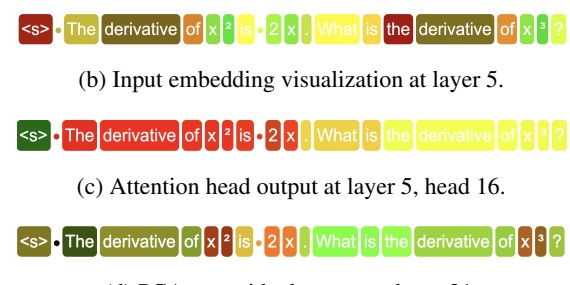

(b) Input embedding visualization at layer 5.

(c) Attention head output at layer 5, head 16.

(d) PCA on residual stream at layer 21.

Figure 16: Visualization of the 'catch, tag, release' mechanism on a Chain of Thought (CoT) prompt on the PHI-3 MEDIUM model.

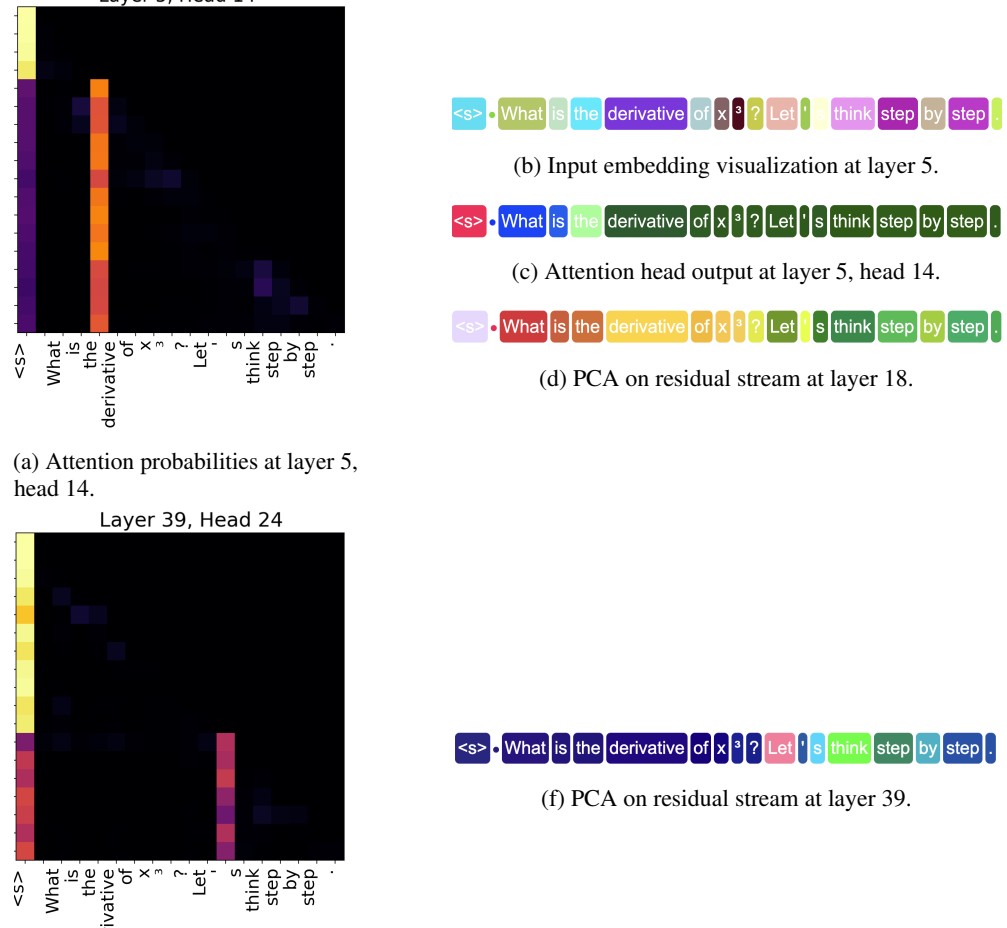

(a) Attention probabilities at layer 5, head 14.

(b) Input embedding visualization at layer 5.

(c) Attention head output at layer 5, head 14.

(d) PCA on residual stream at layer 18.

(e) Attention probabilities at layer 39, head 24.

(f) PCA on residual stream at layer 39.

Figure 17: Visualization of the 'catch, tag, release' mechanism on a Chain of Thought (CoT) prompt on the PHI-3 MEDIUM model. Later layers reveal an attention sink forming on "Let's think step by step", with the output embedding retaining features from tagged value embeddings.

## B Taxonomy of Sinks

To explore which tokens commonly exhibit as sinks, we count the frequency of tokens that appear as sinks and measure the average explained variance of each token when it appears as a sink. To this end, we analyzed 200 prompts consisting of 1024 tokens from the QuALITY dataset [Pang et al., 2022], using both the base and reasoning-tuned versions of the QWEN 14B model. The results are presented in Table 3 below.

| Model | Token | Avg. Variance Explained | Frequency |
|---|---|---|---|
| | [FIRST] | 0.2798 | 328 487 |
| | . | 0.2503 | 5323 |
| | , | 0.1998 | 3807 |
| | Ġthe | 0.2334 | 3333 |
| QWEN 2.5 14B | ĠI | 0.2279 | 3283 |
| | Ġ" | 0.2440 | 3203 |
| | Ġto | 0.1902 | 2603 |
| | Ġbegan | 0.2032 | 2592 |
| | ĠStark | 0.1804 | 2512 |
| | ," | 0.2156 | 2253 |
| | [FIRST] | 0.2994 | 307 600 |
| | THE | 0.3087 | 50 248 |
| | The | 0.3005 | 47 535 |
| | Doctor | 0.3032 | 26 928 |
| | SP | 0.2967 | 24 718 |
| DEEPSEEK QWEN 14B | MON | 0.3144 | 23 680 |
| | CAP | 0.3020 | 22 832 |
| | IT | 0.2880 | 22 416 |
| | GR | 0.3019 | 21 915 |
| | IMAGE | 0.2961 | 20 272 |

Table 3: The ten most frequent attention sink tokens for the QWEN 2.5 14B and DEEPSEEK QWEN 14B models. For each token, we report both its frequency of occurrence as an attention sink and the average variance it explains. The token [FIRST] denotes the first token of a sequence (which may vary across sequences).

### B.1 Discussion: Impact of Reasoning Distillation

In base models such as QWEN 2.5 14B, sink formation predominantly occurs around function words (e.g., Ġthe, Ġto, ĠI) and punctuation (such as . or ,), likely due to their high frequency and syntactic roles. In contrast, the DEEPSEEK QWEN 14B reasoning-tuned model forms attention sinks around semantically significant or task-structuring tokens, such as:

1. *IMAGE* suggests segmentation for multimodal inputs, grouping the token with subsequent image-related descriptions.

2. *Doctor* likely marks named entities relevant for reasoning over domain-specific content.

3. Tokens like *MON*, *CAP*, *IT*, and *GR* are plausible abbreviations or categorical labels (e.g. weekdays, captions, grades) that suggest segmentation for structured data.

4. *SP* may represent speaker or section markers, important in multi-step or dialog-based reasoning.

These sink tokens not only appear with high frequency but also explain a substantial portion of the variance, indicating their importance in structuring the model's internal representation. The presence of such tokens, absent in the base model's sink list, suggests that reasoning fine-tuning reorients the attention sink mechanism from syntactic attractors to semantically meaningful or task-specific units.

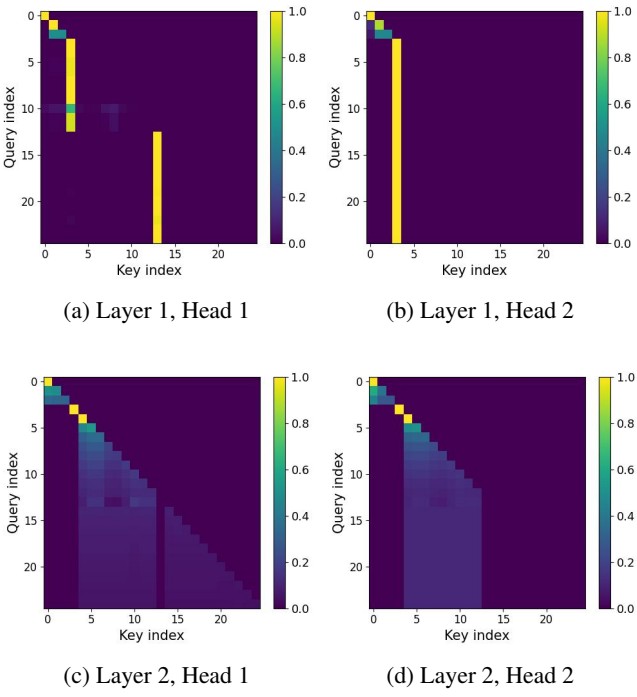

(a) Layer 1, Head 1  (b) Layer 1, Head 2

(c) Layer 2, Head 1  (d) Layer 2, Head 2

Figure 18: Attention probability maps for each head and layer. Each attention head is dedicated to solving a specific portion of the task, which in the first layer, requires the different attention heads to form distinct sinks and tags that the model uses constructively to create the desired probability distributions in the second layer.

## C  Extending the Theoretical Model

The theoretical model presented in Section 7 shows why a single tag and a single attention head suffice in the basic case. A natural extension, however, is to consider a model where:

1. multiple attention sinks are required, and
2. multiple attention heads are employed.

We generalize our theoretical framework to capture these aspects and provide empirical evidence that this extension indeed gives rise to the aforementioned features in the solution.

**Task:**  We modified the averaging task by introducing a second special token, [SEP2], which is inserted at a random position following the [SEP] token. The task is to compute the **sum** of two quantities:

1. the average of the numbers that appear after [SEP], and
2. the **average of the numbers that appear between** [SEP] **and** [SEP2].

**Embeddings:**  We extended the embedding space by one dimension, such that number tokens are embedded as $[x, -1, -1]$. The embeddings for [SEP] and [SEP2] are optimized during training.

**Model:**  We used a two-layer attention-only transformer with:

- Two attention heads per layer with a head dimension of 3,
- A learned output projection $\boldsymbol{W}_O$ in each layer mapping the concatenated heads back to the embedding space.

Depicted in Figure 18 above are the visualizations of the attention weights for the different attention heads for a given sequence where [SEP] lands on token 3 and [SEP2] lands on token 12.

# D    Separability of Tags

A central question in our analysis is whether the model organizes tags within a well-structured and separable subspace of its representation space. If such a subspace exists, it would suggest that tags interact in systematic and potentially predictable ways, rather than interfering with other embedding dimensions. To investigate this, we address two questions:

1. How do different tags interact with one another and do their effects combine additively or interfere with one another?

2. How cleanly can the tag-related components be disentangled from the underlying embeddings?

## D.1    Interaction Between Tags

We measure the cosine similarity between tags across all layers and heads by mapping the values of the tag tokens back into the residual stream.

For a given attention head $h \in \{1, \ldots, H\}$, denote the value of a tag token to be $\boldsymbol{v}_{\text{tag}} \in \mathbb{R}^{d_v}$.

The output projection matrix $\boldsymbol{W}_O \in \mathbb{R}^{d_{\text{model}} \times (H \cdot d_v)}$ acts on the concatenation of all heads. To place $\boldsymbol{v}_{\text{tag}}$ in the correct block of this concatenated space, we embed it via the Kronecker product with a standard basis vector:

$$\tilde{\boldsymbol{v}}_{\text{tag}} \;=\; \mathbf{e}_h \otimes \boldsymbol{v}_{\text{tag}} \quad \in \mathbb{R}^{H \cdot d_v},$$

where $\mathbf{e}_h \in \mathbb{R}^H$ is the one-hot vector with a 1 in the $h$-th position. This construction is equivalent to inserting $\boldsymbol{v}_{\text{tag}}$ into the block corresponding to head $h$ and padding with zeros elsewhere.

We then map the padded vector back into the residual stream as:

$$\hat{\boldsymbol{v}}_{\text{tag}} \;=\; \boldsymbol{W}_O \, \tilde{\boldsymbol{v}}_{\text{tag}} \quad \in \mathbb{R}^{d_{\text{model}}}.$$

Finally, given two tags $a, b$ (possibly from different layers or heads), their similarity in the residual stream is defined as the cosine similarity:

$$\cos(\hat{\boldsymbol{v}}_a, \hat{\boldsymbol{v}}_b) = \frac{\langle \hat{\boldsymbol{v}}_a, \hat{\boldsymbol{v}}_b \rangle}{\|\hat{\boldsymbol{v}}_a\| \, \|\hat{\boldsymbol{v}}_b\|}.$$

The results are shown in Figure 19 below.

## D.2    Interaction Between Tags and Embeddings

We measure the average cosine similarity between the tag component of each token's representation and the embeddings to which it is added in the residual stream.

For a token position $t$ and each head $h = 1, \ldots, H$, define the per-head tag component:

$$\boldsymbol{z}_{\text{tag},t}^{(h)} \;=\; \sum_{k=1}^{T} \mathbf{1}_{[\alpha_k^{(h)} > \epsilon]} \, \boldsymbol{A}_{t,k}^{(h)} \, \boldsymbol{V}_{k,:}^{(h)}.$$

Concatenate the per-head tag components into the head-concatenated vector:

$$\tilde{\boldsymbol{z}}_{\text{tag},t} \;=\; \begin{bmatrix} \boldsymbol{z}_{\text{tag},t}^{(1)} \\ \vdots \\ \boldsymbol{z}_{\text{tag},t}^{(H)} \end{bmatrix} \in \mathbb{R}^{H d_v}.$$

Map this concatenated vector back into the residual stream using the output projection $\boldsymbol{W}_O$:

$$\hat{\boldsymbol{z}}_{\text{tag},t} \;=\; \boldsymbol{W}_O \, \tilde{\boldsymbol{z}}_{\text{tag},t} \;\in\; \mathbb{R}^{d_{\text{model}}}.$$

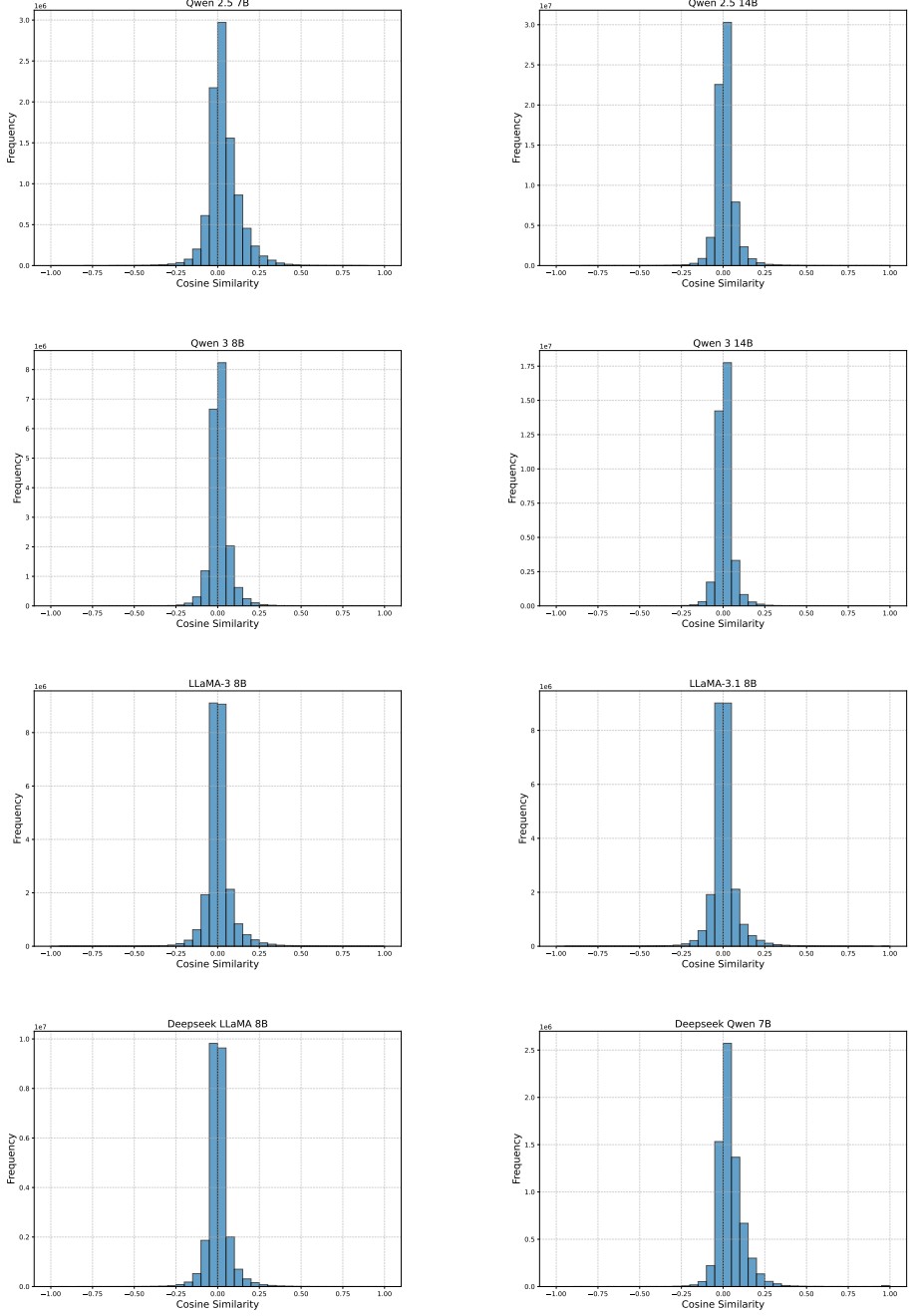

Figure 19: **Cosine Similarity Between Tags.** Histograms of the cosine similarity between distinct tags across all layers and heads. The tags typically exhibit very small cosine similarity implying that they are close to orthogonal and do not interfere with one another.

The cosine similarity between the mapped tag component and the attention layer input $\boldsymbol{x}_t$ (i.e. the embedding to which the attention output is added) is:

$$\cos\left(\hat{\boldsymbol{z}}_{\text{tag},t}, \boldsymbol{x}_t\right) \;=\; \frac{\langle \hat{\boldsymbol{z}}_{\text{tag},t}, \boldsymbol{x}_t \rangle}{\|\hat{\boldsymbol{z}}_{\text{tag},t}\| \, \|\boldsymbol{x}_t\|}.$$

Table 4 below depicts the average of the cosine similarities over 200 prompts consisting of 1024 tokens for various models.

| Model | Cosine Similarity |
|---|---|
| DEEPSEEK LLAMA 8B | -0.0674 |
| DEEPSEEK QWEN 7B | -0.1651 |
| DEEPSEEK QWEN 14B | -0.1449 |
| LLAMA 3 8B | 0.0238 |
| LLAMA 3.1 8B | 0.0099 |
| QWEN 3 8B | -0.0868 |
| QWEN 3 14B | -0.1290 |
| QWEN 2.5 8B | -0.1738 |
| QWEN 2.5 7B | -0.1581 |
| QWEN 2.5 14B | -0.1444 |

Table 4: **Cosine similarity between tags and embeddings.** The average cosine similarity is low across all models, especially in the LLaMA models, which suggests that the tag subspace remains largely orthogonal to the native token representations.

# E   Proof of Theorem 7.1

We will use the auxiliary notation:

$$\boldsymbol{A}^1 = \text{softmax}(\boldsymbol{E}\boldsymbol{E}^\top)$$

$$\boldsymbol{A}^2 = \text{softmax}(\boldsymbol{H}\boldsymbol{W}_Q^2\boldsymbol{W}_K^2\boldsymbol{H}^\top),$$

## E.1   Proof of Theorem, Part 1: 'Catch, Tag, Release'

The first part of the proof focuses on the first attention layer.

**Catch Mechanism**   The attention weight when token $\boldsymbol{x}_i$, for $i > t$ is attending to [SEP] is:

$$
\begin{aligned}
\boldsymbol{A}_{i,t}^1 &= \frac{\exp(\boldsymbol{e}_i^\top \boldsymbol{e}_t)}{\sum_{k=1}^{i} \exp(\boldsymbol{e}_i^\top \boldsymbol{e}_k)} \\
&= \frac{\exp(\boldsymbol{e}_i^\top \boldsymbol{e}_t)}{\exp(\boldsymbol{e}_i^\top \boldsymbol{e}_t) + \sum_{k=1,k\neq t}^{i} \exp(\boldsymbol{e}_i^\top \boldsymbol{e}_k)} \\
&= \frac{\exp(x_i s_{\text{num}} + s_{\text{tag}})}{\exp(x_i s_{\text{num}} + s_{\text{tag}}) + \sum_{k=1,k\neq t}^{i} \exp(x_i x_k + 1)} \\
&= \frac{\exp(s_{\text{tag}})}{\exp(s_{\text{tag}}) + \sum_{k=1,k\neq t}^{i} \exp(x_i x_k + 1)} \xrightarrow{s_{\text{tag}}\to\infty} 1.
\end{aligned}
$$

Furthermore, for $i > t$ and $j \neq t$:

$$\boldsymbol{A}_{i,j}^1 = \frac{\exp(1 + x_i x_j)}{\exp(s_{\text{tag}}) + \sum_{\substack{k=1 \\ k\neq j}}^{i} \exp(1 + x_i x_k)} \in \mathcal{O}(e^{-s_{\text{tag}}}) \tag{5}$$

and therefore:

$$A^1_{i,j} \xrightarrow{s_{\text{tag}} \to \infty} 0 \quad \text{for} \quad j \neq t.$$

Thus, [SEP] acts as an attention sink, where all tokens $x_i$ for $i > t$, as well as the [SEP] token itself, attend exclusively to the [SEP] token. As a result, the attention of all tokens $x_i$ for $i > t$ and the [SEP] token have been caught.

**Tag Mechanism**    The attention output of the $i$-th token for $i \geq t$ is:

$$\text{Attention}(e_i^\top, E, EW_V^1)$$

$$= \sum_{j=1}^{i} A^1_{i,j} e_j^\top W_V^1$$

$$= A^1_{i,t} e_t^\top W_V^1 + \sum_{j=1, j \neq t}^{i} A^1_{i,j} e_j^\top W_V^1 \tag{6}$$

$$= A^1_{i,t} \begin{bmatrix} 0 & s_{\text{tag}} \end{bmatrix} + \sum_{j=1, j \neq t}^{i} A^1_{i,j} e_j^\top W_V^1 \xrightarrow{s_{\text{tag}} \to \infty} \begin{bmatrix} 0 & \infty \end{bmatrix}.$$

Recall that the second coordinate of the embeddings represents the tag. The limit above implies that a tag has been created for all tokens $x_i$, for $i > t$.

After adding the tag to the residual stream, we obtain for $i > t$:

$$h_i = e_i + \text{Attention}(e_i^\top, E, EW_V^1)^\top \xrightarrow{s_{\text{tag}} \to \infty} \begin{bmatrix} x_i \\ \infty \end{bmatrix} \tag{7}$$

The above implies that all tokens, $x_i$, for $i > t$ have now been tagged.

**Release Mechanism**    The tagged tokens have now been released into the residual stream. As we will show in the next subsection, the tags will be leveraged by the second attention layer to generate the desired averaging mechanism.

### E.2  Proof of Theorem, Part 2: Leveraging the Tags

The second part of the proof focuses on the second attention layer.

The attention weight when token $x_T$ is attending to token $x_j$ is given by:

$$A^2_{T,j} = \frac{\exp(h_T^\top W_Q^2 W_K^2 h_j)}{\sum_{k=1}^{T} \exp(h_T^\top W_Q^2 W_K^2 h_k)}.$$

Dividing the numerator and denominator by $\exp(h_T^\top W_Q^2 W_K^2 h_T)$, the expression becomes:

$$A^2_{T,j} = \frac{\exp(h_T^\top W_Q^2 W_K^2 h_j - h_T^\top W_Q^2 W_K^2 h_T)}{\sum_{k=1}^{T} \exp(h_T^\top W_Q^2 W_K^2 h_k - h_T^\top W_Q^2 W_K^2 h_T)}$$

$$= \frac{\exp(h_T^\top W_Q^2 W_K^2 (h_j - h_T))}{\sum_{k=1}^{T} \exp(h_T^\top W_Q^2 W_K^2 (h_k - h_T))}. \tag{8}$$

Consider the term inside the exponent:

$$h_T^\top W_Q^2 W_K^2 (h_j - h_T).$$

There are three cases for $j$, depending on whether it is less than, equal to, or greater than $t$. In the next subsection, we prove that

**Case 1:**  For $j < t$ (non-tagged tokens),

$$\exp(h_T^\top W_Q^2 W_K^2 (h_j - h_T)) \xrightarrow{s_{\text{tag}} \to \infty} 0.$$

**Case 2:** For $j = t$ (sink token),

$$\exp(\boldsymbol{h}_T^\top \boldsymbol{W}_Q^2 \boldsymbol{W}_K^2 (\boldsymbol{h}_t - \boldsymbol{h}_T)) \xrightarrow{s_{\text{tag}} \to \infty} 0.$$

**Case 3:** For $j > t$ (tagged tokens),

$$\exp(\boldsymbol{h}_T^\top \boldsymbol{W}_Q^2 \boldsymbol{W}_K^2 (\boldsymbol{h}_j - \boldsymbol{h}_T)) \xrightarrow{s_{\text{tag}} \to \infty} 1.$$

Combining all three cases together with Equation (8) leads to:

$$\lim_{s_{\text{tag}} \to \infty} \boldsymbol{A}_{T,j}^2 = \begin{cases} 0, & j < t \\ 0, & j = t \\ \frac{1}{T-t}, & j > t \end{cases}$$

The above shows how the tags have been leveraged to arrive at a uniform distribution over the desired tokens. Then using this, Equation (7), and that $\boldsymbol{W}_V^2 = \begin{bmatrix} 1 \\ 0 \end{bmatrix}$, we can conclude that:

$$f_\theta(\boldsymbol{x}) \xrightarrow{s_{\text{tag}} \to \infty} \frac{1}{T-t} \sum_{i=1}^T x_i.$$

### E.3  Proof of Theorem, Part 3: Proving the Three Cases

Combining Equation (6) and Equation (7), we get for all $1 \le i \le T$:

$$\boldsymbol{h}_i^\top = \boldsymbol{e}_i^\top + \boldsymbol{A}_{i,t}^1 \boldsymbol{e}_t^\top \boldsymbol{W}_V^1 + \sum_{\substack{k=1 \\ k \ne t}}^i \boldsymbol{A}_{i,k}^1 \boldsymbol{e}_k^\top \boldsymbol{W}_V^1,$$

and specifically:

$$\boldsymbol{h}_T^\top = \boldsymbol{e}_T^\top + \boldsymbol{A}_{T,t}^1 \boldsymbol{e}_t^\top \boldsymbol{W}_V^1 + \underbrace{\sum_{\substack{k=1 \\ k \ne t}}^T \boldsymbol{A}_{T,k}^1 \boldsymbol{e}_k^\top \boldsymbol{W}_V^1}_{\mathcal{O}(e^{-s_{\text{tag}}})}$$

$$\implies \boldsymbol{h}_T = \begin{bmatrix} x_T \\ -1 \end{bmatrix} + \boldsymbol{A}_{T,t}^1 \begin{bmatrix} 0 \\ s_{\text{tag}} \end{bmatrix} + \mathcal{O}(e^{-s_{\text{tag}}}).$$

Using the two equations above, we get for all $1 \le j \le T$:

$$\boldsymbol{h}_j^\top - \boldsymbol{h}_T^\top = \boldsymbol{e}_j^\top - \boldsymbol{e}_T^\top + (\boldsymbol{A}_{j,t}^1 - \boldsymbol{A}_{T,t}^1) \boldsymbol{e}_t^\top \boldsymbol{W}_V^1$$

$$+ \underbrace{\sum_{\substack{k=1 \\ k \ne t}}^j \boldsymbol{A}_{j,k}^1 \boldsymbol{e}_k^\top \boldsymbol{W}_V^1}_{\mathcal{O}(1) \text{ if } j < t, \text{ otherwise } \mathcal{O}(e^{-s_{\text{tag}}})} - \underbrace{\sum_{\substack{k=1 \\ k \ne t}}^T \boldsymbol{A}_{T,k}^1 \boldsymbol{e}_k^\top \boldsymbol{W}_V^1}_{\mathcal{O}(e^{-s_{\text{tag}}})},$$

where the order of magnitude of the last two terms is given by Equation (5).

Additionally, we will use the fact that for all $j \ge t$:

$$\lim_{s_{\text{tag}} \to \infty} d \cdot \boldsymbol{A}_{T,t}^1 (\boldsymbol{A}_{j,t}^1 - \boldsymbol{A}_{T,t}^1) s_{\text{tag}}^2 = 0,$$

which we show in Appendix E.4.

**Case 1:** For $j < t$ (non-tagged tokens),

$$\boldsymbol{h}_T^\top \boldsymbol{W}_Q^2 \boldsymbol{W}_K^2 (\boldsymbol{h}_j - \boldsymbol{h}_T)$$

$$= \boldsymbol{h}_T^\top \boldsymbol{W}_Q^2 \boldsymbol{W}_K^2 \left( \begin{bmatrix} x_j - x_T \\ 0 \end{bmatrix} - \boldsymbol{A}_{T,t}^1 \begin{bmatrix} 0 \\ s_{\text{tag}} \end{bmatrix} + \mathcal{O}(1) \right)$$

$$= \boldsymbol{h}_T^\top \left( -\boldsymbol{A}_{T,t}^1 \begin{bmatrix} b \cdot s_{\text{tag}} \\ d \cdot s_{\text{tag}} \end{bmatrix} + \mathcal{O}(1) \right)$$

$$= \left( \begin{bmatrix} x_T \\ -1 \end{bmatrix} + \boldsymbol{A}_{T,t}^1 \begin{bmatrix} 0 \\ s_{\text{tag}} \end{bmatrix} + \mathcal{O}(e^{-s_{\text{tag}}}) \right)^\top$$

$$\left( -\boldsymbol{A}_{T,t}^1 \begin{bmatrix} b \cdot s_{\text{tag}} \\ d \cdot s_{\text{tag}} \end{bmatrix} + \mathcal{O}(1) \right)$$

$$= -(\boldsymbol{A}_{T,t}^1)^2 \cdot d \cdot s_{\text{tag}}^2 + \mathcal{O}(s_{\text{tag}})$$

$$\xrightarrow{s_{\text{tag}} \to \infty} -\infty, \text{ since } d > 0.$$

**Case 2:** For $j = t$ (sink token),

$$\boldsymbol{h}_T^\top \boldsymbol{W}_Q^2 \boldsymbol{W}_K^2 (\boldsymbol{h}_t - \boldsymbol{h}_T)$$

$$= \boldsymbol{h}_T^\top \boldsymbol{W}_Q^2 \boldsymbol{W}_K^2 \left( \begin{bmatrix} 0 - x_T \\ -s_{\text{tag}} + 1 \end{bmatrix} \right.$$

$$\left. + (\boldsymbol{A}_{t,t}^1 - \boldsymbol{A}_{T,t}^1) \begin{bmatrix} 0 \\ s_{\text{tag}} \end{bmatrix} + \mathcal{O}(e^{-s_{\text{tag}}}) \right)$$

$$= \boldsymbol{h}_T^\top \left( \begin{bmatrix} b \cdot (1 - s_{\text{tag}}) \\ d \cdot (1 - s_{\text{tag}}) \end{bmatrix} \right.$$

$$\left. + (\boldsymbol{A}_{t,t}^1 - \boldsymbol{A}_{T,t}^1) \begin{bmatrix} b \cdot s_{\text{tag}} \\ d \cdot s_{\text{tag}} \end{bmatrix} + \mathcal{O}(e^{-s_{\text{tag}}}) \right)$$

$$= -\boldsymbol{A}_{T,t}^1 \cdot d \cdot s_{\text{tag}}^2$$

$$+ \underbrace{d \cdot \boldsymbol{A}_{T,t}^1 (\boldsymbol{A}_{t,t}^1 - \boldsymbol{A}_{T,t}^1) s_{\text{tag}}^2}_{\xrightarrow{s_{\text{tag}} \to \infty} 0} + \mathcal{O}(s_{\text{tag}})$$

$$\xrightarrow{s_{\text{tag}} \to \infty} -\infty.$$

**Case 3:** For $j > t$ (tagged tokens),

$$\boldsymbol{h}_T^\top \boldsymbol{W}_Q^2 \boldsymbol{W}_K^2 (\boldsymbol{h}_j - \boldsymbol{h}_T)$$

$$= \boldsymbol{h}_T^\top \boldsymbol{W}_Q^2 \boldsymbol{W}_K^2 \left( \begin{bmatrix} x_j - x_T \\ 0 \end{bmatrix} + \right.$$

$$\left. + (\boldsymbol{A}_{j,t}^1 - \boldsymbol{A}_{T,t}^1) \begin{bmatrix} 0 \\ s_{\text{tag}} \end{bmatrix} + \mathcal{O}(e^{-s_{\text{tag}}}) \right)$$

$$= \boldsymbol{h}_T^\top \left( (\boldsymbol{A}_{j,t}^1 - \boldsymbol{A}_{T,t}^1) \begin{bmatrix} b \cdot s_{\text{tag}} \\ d \cdot s_{\text{tag}} \end{bmatrix} + \mathcal{O}(e^{-s_{\text{tag}}}) \right)$$

$$= (\boldsymbol{A}_{j,t}^1 - \boldsymbol{A}_{T,t}^1)(x_T \cdot b \cdot s_{\text{tag}} - d \cdot s_{\text{tag}})$$

$$+ d \cdot \boldsymbol{A}_{T,t}^1 (\boldsymbol{A}_{j,t}^1 - \boldsymbol{A}_{T,t}^1) s_{\text{tag}}^2$$

$$+ \mathcal{O}(s_{\text{tag}} e^{-s_{\text{tag}}})$$

$$\xrightarrow{s_{\text{tag}} \to \infty} 0,$$

where $\lim_{s_{\text{tag}} \to \infty} (\boldsymbol{A}_{j,t}^1 - \boldsymbol{A}_{T,T}^1) s_{\text{tag}} = 0$ is proved in Appendix E.4 below.

## E.4 Limit Proof

We first show that:

$$\lim_{s_{\text{tag}} \to \infty} d \cdot \boldsymbol{A}_{T,t}^1 (\boldsymbol{A}_{t,t}^1 - \boldsymbol{A}_{T,t}^1) s_{\text{tag}}^2 = 0$$

By definition:

$$\boldsymbol{A}_{T,t}^1 = \frac{\exp(s_{\text{tag}})}{\exp(s_{\text{tag}}) + \sum_{k=1, k \neq t}^{T} \exp(x_T x_k + 1)} = \frac{1}{1 + \sum_{k=1, k \neq t}^{T} \exp(x_T x_k + 1 - s_{\text{tag}})}$$

and that:

$$\boldsymbol{A}_{t,t}^1 = \frac{\exp(s_{\text{tag}}^2)}{\exp(s_{\text{tag}}^2) + (t-1) \exp(s_{\text{tag}})} = \frac{1}{1 + (t-1) \exp(s_{\text{tag}} - s_{\text{tag}}^2)}.$$

Thus,

$$\boldsymbol{A}_{t,t}^1 - \boldsymbol{A}_{T,t}^1 = \frac{\sum_{k=1, k \neq t}^{T} \exp(x_T x_k + 1 - s_{\text{tag}}) - (t-1) \exp(s_{\text{tag}} - s_{\text{tag}}^2)}{\left(1 + (t-1) \exp(s_{\text{tag}} - s_{\text{tag}}^2)\right) \cdot \left(1 + \sum_{k=1, k \neq t}^{T} \exp(x_T x_k + 1 - s_{\text{tag}})\right)}$$

Then:

$$\lim_{s_{\text{tag}} \to \infty} s_{\text{tag}}^2 (\boldsymbol{A}_{t,t}^1 - \boldsymbol{A}_{T,t}^1)$$

$$= \lim_{s_{\text{tag}} \to \infty} \frac{\sum_{k=1, k \neq t}^{T} s_{\text{tag}}^2 \exp(x_T x_k + 1 - s_{\text{tag}}) - (t-1) s_{\text{tag}}^2 \exp(s_{\text{tag}} - s_{\text{tag}}^2)}{\left(1 + (t-1) \exp(s_{\text{tag}} - s_{\text{tag}}^2)\right) \cdot \left(1 + \sum_{k=1, k \neq t}^{T} \exp(x_T x_k + 1 - s_{\text{tag}})\right)}$$

$$= \frac{0}{1} = 0$$

The rest follows from the fact that $\lim_{s_{\text{tag}} \to \infty} \boldsymbol{A}_{T,t}^1 = 1$ and applying basic limit laws.

We now show similarly that for $j > t$:

$$\lim_{s_{\text{tag}} \to \infty} (\boldsymbol{A}_{j,t}^1 - \boldsymbol{A}_{T,T}^1) s_{\text{tag}} = 0, \qquad \lim_{s_{\text{tag}} \to \infty} (\boldsymbol{A}_{j,t}^1 - \boldsymbol{A}_{T,T}^1) s_{\text{tag}}^2 = 0$$

Observe that for $j > t$:

$$\boldsymbol{A}_{j,t}^1 = \frac{\exp(s_{\text{tag}})}{\exp(s_{\text{tag}}) + \sum_{k=1, k \neq t}^{j} \exp(x_j x_k + 1)} = \frac{1}{1 + \sum_{k=1, k \neq t}^{j} \exp(x_j x_k + 1 - s_{\text{tag}})}$$

Then

$$\lim_{s_{\text{tag}} \to \infty} (\boldsymbol{A}_{j,t}^1 - \boldsymbol{A}_{T,T}^1) s_{\text{tag}}$$

$$= \frac{\sum_{k=1, k \neq t}^{T} s_{\text{tag}} \exp(x_T x_k + 1 - s_{\text{tag}}) - \sum_{k=1, k \neq t}^{j} s_{\text{tag}} \exp(x_j x_k + 1 - s_{\text{tag}})}{\left(1 + \sum_{k=1, k \neq t}^{j} \exp(x_j x_k + 1 - s_{\text{tag}})\right) \cdot \left(1 + \sum_{k=1, k \neq t}^{T} \exp(x_T x_k + 1 - s_{\text{tag}})\right)}$$

$$= \frac{0}{1} = 0$$

and similarly,

$$\lim_{s_{\text{tag}} \to \infty} (\boldsymbol{A}_{j,t}^1 - \boldsymbol{A}_{T,T}^1) s_{\text{tag}}^2$$

$$= \frac{\sum_{k=1, k \neq t}^{T} s_{\text{tag}}^2 \exp(x_T x_k + 1 - s_{\text{tag}}) - \sum_{k=1, k \neq t}^{j} s_{\text{tag}}^2 \exp(x_j x_k + 1 - s_{\text{tag}})}{\left(1 + \sum_{k=1, k \neq t}^{j} \exp(x_j x_k + 1 - s_{\text{tag}})\right) \cdot \left(1 + \sum_{k=1, k \neq t}^{T} \exp(x_T x_k + 1 - s_{\text{tag}})\right)}$$

$$= \frac{0}{1} = 0$$

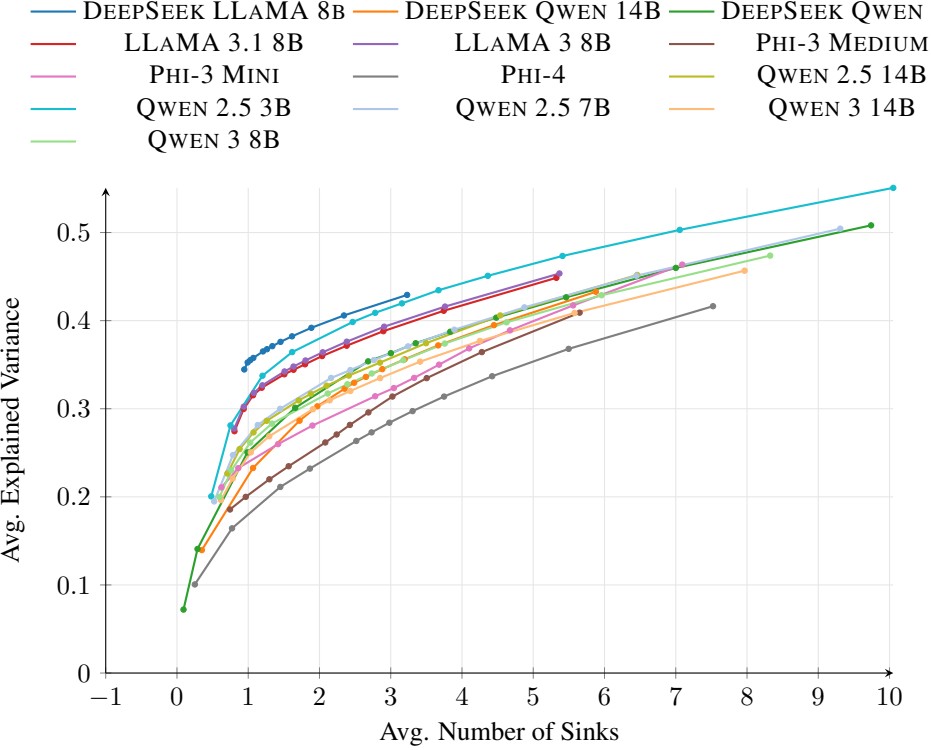

Figure 20: Average explained variance plotted against the average number of sinks for different values of $\varepsilon$. Each marker indicates a specific $\varepsilon$ in the set $\{0.03, 0.04, 0.05, 0.06, 0.07, 0.08, 0.09, 0.1, 0.15, 0.2, 0.3, 0.4\}$.

## F   Sensitivity Analysis of the Sink Threshold

We performed a comprehensive sensitivity analysis by varying the threshold across twelve values from $0.03$ to $0.4$ and evaluating its effect on two key metrics:

1. the average number of identified sinks, and
2. the average variance explained by the tags.

The results are shown in Figure 20 above.

Across all models tested, the resulting curves are typically logarithmic-like: initially, decreasing the threshold adds new sinks and increases explained variance, but this gain saturates beyond a certain point. This suggests that smaller thresholds tend to capture less meaningful or redundant tokens. The table below provides representative values from this analysis to support this observation.

## G   Optimization Details for Section 7.3

We generate a total of 16,384 sequences of length 16 where half are allocated for training while the other half is allocated for evaluation. For each sequence, a random index is generated to place the [SEP]. We minimize the mean squared error loss between the predicted and actual average and employ a cosine annealing learning rate scheduler. We employ an initialization of $s_{\tt tag} = 10$ to aid with convergence. Depicted in Figure 21 are the predicted averages of the trained model versus the actual average on the evaluation set.

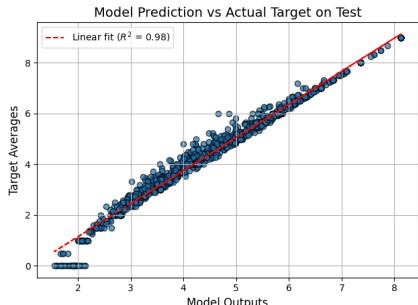

Figure 21: The predicted averages of the trained model versus the actual average on the evaluation set.

## H  Qwen 3 Results

We present quantitative evidence that the QWEN 3 models [Yang et al., 2025], which incorporate QK normalization, exhibit the 'catch, tag, release' mechanism.

### H.1  Sink Count and Variance Explained by Sinks

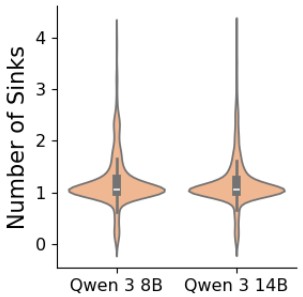

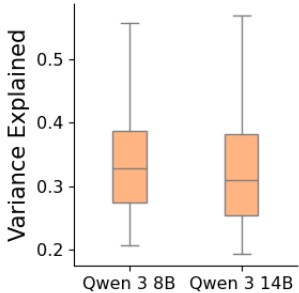

(a) **Attention Sink Counts for Qwen 3.** Counts are computed for each head and their distribution is visualized for each model using violin plots.

(b) **Variance Explained by Tags for Qwen 3.** The metric is computed for each head and summary statistics are shown for each model using box plots.

Figure 22: **Quantitative Measurements for Qwen 3.** Analogous measurements presented in Section 2 for models in the Qwen 3 family. Notably, these models exhibit the 'catch, tag, release' mechanism while having QK normalization.

### H.2  Probe Performance

| Probe | QWEN 3 | |
|---|---|---|
| | **8B** | **14B** |
| $\theta_{\text{tag}}$ | **100%** | **87.0%** |
| $\theta_{\text{no tag}}$ | 50.5% | 64.5% |
| $\theta_{\text{activation}}$ | 56.0% | 82.0% |

Table 5: **Classification Accuracy of Probes for Qwen 3.** The probe $\theta_{\text{tag}}$ is computed from the *tag* component of the activation, $\theta_{\text{no tag}}$ from the *non-tag* component, and $\theta_{\text{activation}}$ from the *full* activation.

## I  Additional Discussion

### I.1  Practical Applications to LLMs

The presence and utility of the 'catch, tag, release' mechanism have important implications for practical use of large language models. Section 4 shows the potential for tags to be used to steer activations [Subramani et al., 2022, Turner et al., 2025] which can be applied for model alignment and safety [Wang and Shu, 2024]. In prior work Yona et al. [2025], attention sinks were linked to repeated token phenomena and jailbreaking, suggesting the potential for the mechanism to be used for both exploit detection and mitigation. In model compression, preserving this mechanism may be critical, as we hypothesize it can be captured within a low-rank subspace of the model's parameters, which must be retained during pruning or quantization to maintain performance [Zhang and Papyan, 2025b, Makni et al., 2025]. Furthermore, recent work, Wang et al. [2025], has identified low-entropy tokens as central to LLM reasoning abilities, likely overlapping with the tag tokens we highlight, indicating this mechanism may play a foundational role in LLM reasoning.

## I.2 Role of the Fully Connected Layers

Empirically, prior work has shown that groups of token representations tend to collapse to low-dimensional subspaces, or even single points, across layers [Geshkovski et al., 2025]. This raises the question: which tokens collapse together, and what determines the target of this collapse? We hypothesize that the 'catch, tag, release' mechanism plays a central role in both the grouping and the collapse target. Specifically, tokens attending to the same attention sink receive a common tag, which defines the direction and destination of their collapse. We further propose that the MLP layers, via a low-rank subspace, progressively drive this collapse across layers by reinforcing the shared tag structure.

# J  Mass-Mean Probe Details

The ([CITY], [COUNTRY]) pairs are sourced from GeoNames [GeoNames, 2025].

## J.1  Layer and Head Information

For Table 1, the layer and attention head for each model that we extracted the probes from are depicted in Table 6 below:

|  | Qwen 2.5 | | | Llama-3 8B | Llama-3.1 8B |
|---|---|---|---|---|---|
|  | 3B | 7B | 14B |  |  |
| Layer | 24 | 16 | 40 | 18 | 18 |
| Head | 4 | 28 | 40 | 21 | 21 |

Table 6: Layer/Attention Head information for results presented in Table 1.

## J.2  Taxonomy of Tag Tokens

For each tag-only probe presented in Table 1, we present the histogram of the tag tokens that emerged during their generation in Figure 23 below.

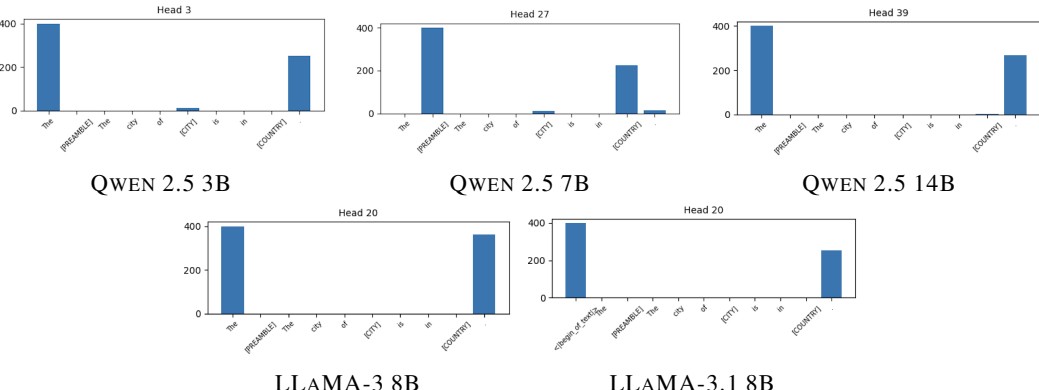

Figure 23: A taxonomy of the tokens that were identified as attention sinks during the generation of the $\theta_{\text{tag}}$ and $\theta_{\text{no tag}}$ probes. [PREAMBLE] is being used to represent all tokens that correspond to the following portion of the prompt: The city of Tokyo is in Japan. This statement is TRUE. The city of Hanoi is in Poland. This statement is: FALSE.

### J.3 Definition of $\theta_{\text{activation}}$

Following Marks and Tegmark [2024], the activation-based probes, $\theta_{\text{activation}}$, are computed from the full activations:

$$z_t = \sum_{k=1}^{T} A_{t,k} V_{k,:}.$$

From these representations, we obtain the class means:

$$\mu^+ = \frac{1}{N_+} \sum_{i \in \text{True}} z_t^{(i)}, \quad \mu^- = \frac{1}{N_-} \sum_{i \in \text{False}} z_t^{(i)}.$$

The within-class covariance matrix, $\Sigma$, is then estimated, and the resulting mass-mean probes are defined as:

$$\theta_{\text{activation}}(z) = \sigma\left(z^\top \Sigma^{-1}(\mu^+ - \mu^-)\right).$$

### J.4 Sentiment Analysis

To explore how the tagging mechanism can potentially capture a broader range of semantic and syntactic information beyond just binary True/False classifications, we extend our experiments to assess whether tags can encode sentiment. Specifically, we construct prompts in the format shown in Figure 24 on the right.

> it's a charming and often affecting journey.
> This statement is: POSITIVE.
> unflinchingly bleak and desperate.
> This statement is: NEGATIVE.
> [SENTENCE]. ⟵——
> This statement is:

Figure 24: Sentiment prompt example.

[SENTENCE] is replaced with various examples from the SST-2 dataset [Socher et al., 2013]. Using the same experimental setup described in Section 4, we evaluated whether tags could capture sentiment polarity by decomposing the activations of the final period token (indicated by the arrow).

The results, presented in Table 7 below, show that tags indeed encode positive and negative sentiment information effectively, supporting the broader applicability of the method to a variety of different semantic tasks.

| Probe | QWEN 2.5 | | | LLAMA-3 | LLAMA-3.1 |
|---|---|---|---|---|---|
| | 3B | 7B | 14B | 8B | 8B |
| $\theta_{\text{tag}}$ | **88.5**% | **90.0**% | **94.0**% | **87.5**% | **86.5**% |
| $\theta_{\text{no tag}}$ | 84.0% | 53.0% | 53.5% | 81.5% | 64.0% |
| $\theta_{\text{activation}}$ | 86.5% | 80.0% | 90.5% | 88.0% | 83.0% |

Table 7: **Classification Accuracy of Probes.** Comparison of the three probe sets on positive/negative sentiment analysis.

## K Limitations

### K.1 Theoretical Model

While our theoretical result provides a constructive proof that the 'catch, tag, release' mechanism can emerge in transformer architectures, it comes with limitations detailed below.

Theorem 7.1 does not prove that the 'catch, tag release' mechanism is necessary. There may exist alternative solutions that perform the task without attention sinks or outlier features. Consequently, the theoretical result should be viewed as illustrative. Although we provide empirical support suggesting that similar dynamics arise in trained models, we do not prove convergence to our specific construction under training.

The theoretical model operates under highly constrained conditions: a two-layer transformer solving a sequence averaging problem. While this setting is valuable for analytical tractability, it is removed from the complexity of real-world language modeling tasks.

Furthermore, depite the theoretical model being simple, the optimization process does not always reach a low-loss solution. We found that convergence depends on how the $s_{\texttt{tag}}$ parameter is initialized, with larger initial values generally leading to more consistent success. Table 8 below shows how often the model achieved a high fit ($R^2 > 0.95$) across 10 runs for different starting values of $s_{\texttt{tag}}$:

| $s_{\text{tag}}$ initialization | Success Rate |
| --- | --- |
| 10 | 4/10 |
| 6 | 3/10 |
| 1 | 0/10 |

Table 8: Success rate, measured by $R^2 > 0.95$, across different $s_{\text{tag}}$ initializations.

### K.2 Connection with Reasoning

While our findings in Section 5 strongly suggest a structural role for sinks in reasoning, they do not yet establish a causal link between the two. This remains an important direction for future work.

### K.3 Threshold-Based Metric

The threshold-based metric used to identify attention sinks does not guarantee full coverage of all such tokens. Our choice of a $\varepsilon = 0.2$ threshold, while consistent with prior studies, is not necessarily optimal. As discussed in Gu et al. [2024], there is currently no principled method for determining this threshold. A sensitivity analysis for this threshold is provided in Appendix F.

## L  Compute Resources

All experiments involving LLMs were executed utilizing a single NVIDIA A40 with 48GB of GPU memory. This includes the experiments used to generate Figures 2, 3, 5, 6, Table 1, and visualizations provided in Appendix A. Each of the experiments in Sections 3, 5, 6 took approximately 40 minutes per model while the visualizations in Section 2 took roughly 60 minutes per model totalling for roughly 40 hours. Roughly 30 additional hours were utilized for preliminary experiments that did not reach the final paper.

## M  Models and Implementation

A comprehensive list of all the models used in our empirical study is found below:

- Phi-3 Mini and Phi-3 Medium [Abdin et al., 2024a]
- Phi-4 [Abdin et al., 2024b]
- Qwen 2.5 3B, 7B, 14B, 32B [Yang et al., 2024]
- LLaMA-3 8B, LLaMA 3.1 8B [Grattafiori et al., 2024]
- Mistral 7B v0.3 [Jiang et al., 2023]
- Deepseek-R1 LLaMA 8B, Deepseek-R1 Qwen 14B, Deepseek Math 7B [DeepSeek-AI, 2025]
- Gemma 2 9B [Gemma Team et al., 2024]
- Gemma 3 4B, 12B [Gemma Team et al., 2025]
- Qwen 3 4B, 8B, 14B [Yang et al., 2025]

We utilize the Transformers library by Huggingface [Wolf et al., 2020] to run all LLM experiments.

