# OpenReview forum: "Attention Sinks: A 'Catch, Tag, Release' Mechanism for Embeddings"
_NeurIPS.cc/2025/Conference — NeurIPS 2025 poster_

### Official Review · Reviewer_KjwZ · 2025-07-02

**Clarity:** 2
**Significance:** 3
**Originality:** 3
**Rating:** 5
**Confidence:** 4

**Summary:**

The paper discusses a potential reason why "attention sinks" (i.e., attention patterns where all attention concentrates on the first token) are ubiquitous in large language models (LLMs). It introduces a linear-algebraic mechanism which attention sinks execute in LLMs, which is labeled as "catch, tag, release" --- each attention head experiencing an attention sink will endow the input sequence with a component in a distinctive direction in embedding space given by the span of the value states corresponding to the sink tokens. Empirical evidence on both completely synthetic data and more realistic data show that attention sinks introduce a tag, that these tags explain much of the empirical variance of the outputs of the heads, that they have interpretable semantic meaning in controlled synthetic setups, that they occur more in reasoning-distilled models than only-pretrained models, and that attention sinks occur even when QK normalization is applied.

**Questions:**

See above, but to summarize:
- What is the role of multiple attention heads in practice and in the synthetic setups considered in the work?
- What is the role of other non-attention components in the network (MLP, layer norms, etc) in this mechanism?
- Are there any potential applications of this mechanism to the practice of LLMs (capabilities, interpretability, etc)?

**Ethical Concerns:**

["NO or VERY MINOR ethics concerns only"]

**Final Justification:**

I initially recommended borderline rejection, but now recommend acceptance. The main weakness for me in the work was the case of multiple attention heads, which was previously ignored, but the authors did lots of leg work in the rebuttal to address this point, in my opinion making it a better paper. The other unresolved issues (in my opinion this is a lack of theory/synthetic experiments for the multi-head case) are relatively minor. The paper does seem to have some minor technical/presentation issues, from looking at other reviews, but I think the overall focus, direction, and results are quite solid.

**Limitations:**

Yes (Appendix B and the checklist)

**Paper Formatting Concerns:**

no concerns

**Quality:**

3

**Strengths And Weaknesses:**

Strengths:
- The overall mechanism of each attention head to influence the output is clear and succinct.
- The experimental setup using large models is reasonable for many of the tasks, and may be sufficient to demonstrate the claims (modulo some ambiguity, see below).
- The synthetic experimental setup is unambiguous, the claim is cleanly proved, and the setting may be valuable for future work (potentially in a more complex form).

Weaknesses:
- The writing is not very clear; there is a large ambiguity in the text that seems to be glossed over. Namely: what is the role of multiple attention heads? In particular, for a given input sequence, many attention heads are sinks, not just one. How do the tags interact with each other? Is there constructive or destructive interference (or something more complicated), or are all the "tags" orthogonal? It is especially confusing since in Figure 2 the tokens _later in the network_ seem to adhere strongly to the tags produced by the single discussed head earlier in the network, whereas multiple heads surely were attention sinks. It would be great to clarify these as it would strengthen the core contribution of the work.
- There is little-to-no discussion of applications to the practice of LLMs. The main contributions are in clarifying the different mechanisms proposed in prior work. Even though this work is obviously not applied in nature, it would be good to discuss potential applications of the discussion, e.g. for large-scale interpretability or model surgery.
- The theoretical analysis of the completely synthetic setup is for a 2 layer iterated attention network with one attention head in each layer, not necessarily a transformer. This is understandable because analyzing transformer dynamics is very challenging. Still it would be better to discuss the role of the MLP for instance.

---

> ### Author Rebuttal · Authors · 2025-07-30
>
> We are grateful for the reviewer's thoughtful feedback highlighting areas that warrant further discussion and clarification. We address these concerns below.
>
> ---
>
> ## Namely: what is the role of multiple attention heads? In particular, for a given input sequence, many attention heads are sinks, not just one.
>
> Thank you for this insightful question. **Our findings indicate that each attention head implements an independent instance of the catch, tag, release mechanism**. Concretely, each head selects its own attention sink, captures a distinct subsequence of tokens, and injects a unique tag direction into those tokens’ representations via its value vector (potentially repeating this process for multiple sinks). In this sense, the sink location, the span of caught tokens, and the resulting tag are all specific to the head.
> While some heads may form sinks over overlapping regions, for instance, multiple heads attending to the BOS token, it is also common for sinks to appear at different positions, capturing different subsequences. This enables the model to segment and annotate the input in parallel, with each head providing a distinct structural or semantic signal. To illustrate this more concretely, **the revised manuscript now includes a new figure showing that for a single prompt, two different heads induce distinct sinks corresponding to two different subsequences of the input**.
>
> ---
>
> ## How do the tags interact with each other? Is there constructive or destructive interference (or something more complicated), or are all the "tags" orthogonal?
>
> Thank you for this excellent question. In Figure 2(c), each of the R, G, and B channels corresponds to a different top principal component of the attention head output. The emergence of clearly separated red and green coloring indicates that tokens tagged by different sinks project strongly onto different principal components. Since principal components are orthogonal by definition, this **visual separation provides concrete evidence that the corresponding tags occupy distinct, non-overlapping directions in representation space.**
>
> While this is only a single example, it is representative of a broader trend we observe across models and prompts: **tags created by different sinks tend to align with orthogonal directions**, suggesting that the model avoids destructive interference by disentangling the subspaces in which tags reside. This supports the interpretation that tags from different sinks are not simply superimposed but are instead composed in a structured, non-interfering manner. To show this, **we have measured the cosine similarity between distinct tags across all layers and heads showing that the tags typically exhibit very small cosine similarity implying that they are close to orthogonal.** This will be displayed in a histogram which we present a summary of in the table below:
>
> | Model | [-1.00, -0.60) | [-0.60, -0.20) | [0.20, -0.10) | [-0.10, -0.05) | [-0.05, 0.00) | [0.00, 0.05) | [0.05, 0.10) | [0.10 to 0.20) | [0.20 to 0.60) | [0.60 to 1.00] |
> | --- | --- | --- | --- | --- | --- | --- | --- | --- | --- | --- |
> | Qwen 2.5 3B | 10 | 4405 | 20005 | 44105 | 135895 | 161200 | 83005 | 61085 | 24765 | 290 |
> | Qwen 2.5 7B | 20 | 7595 | 28110 | 59640 | 221350 | 308105 | 159140 | 133025 | 48125 | 545 |
> | Qwen 2.5 14B | 145 | 19445 | 116280 | 357050 | 2286530 | 3065200 | 799030 | 321755 | 68525 | 1055 |
> | Qwen 3 8B | 30 | 5525 | 40680 | 122205 | 690575 | 853650 | 208925 | 88505 | 21095 | 565 |
> | Llama 3 8B | 185 | 18510 | 85120 | 195675 | 935850 | 933545 | 216330 | 128010 | 53995 | 735 |
> | Llama 3.1 8B | 160 | 17080 | 79440 | 192825 | 914310 | 916240 | 213520 | 121045 | 47350 | 530 |
> | Deepseek Llama 8B | 150 | 14810 | 70610 | 186825 | 985025 | 965665 | 200615 | 100995 | 32755 | 380 |
> | Deepseek Qwen 7B | 60 | 1245 | 6380 | 21075 | 158700 | 288795 | 161495 | 111735 | 26150 | 1335 |
> ---
>
> ---
>
> ## It is especially confusing since in Figure 2 the tokens later in the network seem to adhere strongly to the tags produced by the single discussed head earlier in the network, whereas multiple heads surely were attention sinks.
>
> We appreciate the reviewer’s observation and agree that further clarification is warranted. While Figure 2(a) illustrates the behavior of a single attention head exhibiting a specific sink structure, in this particular example, **other heads in the layer independently converge on the same sink structure**. This results in the tags from multiple heads reinforcing one another and producing a coherent clustering pattern deeper in the network, as shown in Figure 2(d).
>
> To clarify this, we have revised the caption of Figure 2(d) to read:
> >Tagged tokens, created by multiple attention heads that independently exhibit the same sink structure, propagate through the residual stream and cluster in a deeper layer based on their associated sink.
>
> However, we emphasize that this **alignment across heads is not guaranteed and does not always occur**. To illustrate a contrasting case, we have **added a new figure where different heads create distinct sinks, resulting in more heterogeneous tagging across heads**. We hope this comparison clarifies that the example in Figure 2 is representative of one possible, and particularly clean, case rather than a general rule.
>
> ---
>
> ## What is the role of multiple attention heads… in the synthetic setups considered in the work?
>
> We deliberately designed a simple synthetic task that can be solved using a single attention head to isolate and study the catch, tag, release mechanism. A single head is sufficient in this case because it can form one attention sink and tag a single subsequence starting from a specific source token. However, **this setup limits the model to tagging only one subsequence per source token. To tag multiple subsequences that all begin at the same token but end at different positions, multiple attention heads are needed -- each forming its own sink and injecting a distinct tag**. For instance, a more complex variant of the task could require the model to extract and average several subsequences (e.g., tokens 3–7, 3–15, and 3–28, all starting from token 3) and then combine those averages. This would naturally encourage the use of multiple heads, each capturing and processing a different subsequence.
>
> ---
>
> ## There is little-to-no discussion of applications to the practice of LLMs.
>
> We have added the following paragraph discussing various applications of the mechanism:
>
> > The presence and utility of the `catch, tag, and release’ mechanism have important implications for practical use of large language models. Section 4 shows the potential for tags to be used to steer activations [1] which can be applied for model alignment and safety [2]. In prior work [3], attention sinks were linked to repeated token phenomena and jailbreaking, suggesting the potential for the mechanism to be used for both exploit detection and mitigation. In model compression, preserving this mechanism may be critical, as we hypothesize it can be captured within a low-rank subspace of the model’s parameters, which must be retained during pruning or quantization to maintain performance [4,5]. Furthermore, recent work, [6], has identified low-entropy tokens as central to LLM reasoning abilities, likely overlapping with the tag tokens we highlight, indicating this mechanism may play a foundational role in LLM reasoning.
>
> ---
>
> ## What is the role of other non-attention components in the network (MLP, ...) in this mechanism?
>
> We have **added the following paragraph to discuss the potential role of the MLP in the transformer layers in light of the catch, tag, release mechanism**:
>
> > Prior theoretical work studying transformers from a dynamical systems perspective has shown that groups of token representations tend to collapse to low-dimensional subspaces, or even single points, across layers [7]. This raises the question: which tokens collapse together, and what determines the target of this collapse? We hypothesize that the catch, tag, release mechanism plays a central role in both the grouping and the collapse target. Specifically, tokens attending to the same attention sink receive a common tag, which defines the direction and destination of their collapse. We further **propose that the MLP layers progressively drive the collapse across layers by reinforcing the shared tag structure, effectively denoising the token representations toward the shared tag**.
> >
> > We also envision this mechanism being used in practice by researchers and practitioners to better understand and intervene in model behaviour. For example, one could input a prompt, extract the tags assigned to tokens, and highlight, for each tag, the tokens grouped by it. This would reveal how the model segments and jointly processes subsequences of input, offering a transparent window into its internal computation.
>
> ---
>
> We sincerely thank the reviewer once again for highlighting areas of ambiguity and missing discussion in our manuscript. If the reviewer feels that our revisions have sufficiently addressed these concerns, we would be grateful for any reconsideration of the score.
> >
>
> [1] Turner et al. (2023). Steering Language Models With Activation Engineering.
>
> [2] Wang et al. (2023). Trojan Activation Attack: Red‑Teaming Large Language Models using Activation Steering for Safety‑Alignment.
>
> [3] Yona et al. (2025). Interpreting the Repeated Token Phenomenon in Large Language Models.
>
> [4] Zhang et al. (2024). OATS: Outlier-Aware Pruning Through Sparse and Low Rank Decomposition.
>
> [5] Makni et al. (2025). HASSLE-free: A Unified Framework for Sparse plus Low-Rank Matrix Decomposition for LLMs.
>
> [6] Wang et al. (2025). Beyond the 80/20 Rule: High‑Entropy Minority Tokens Drive Effective Reinforcement Learning for LLM Reasoning.
>
> [7] Geshkovski et al. (2023). A mathematical perspective on Transformers.

---

> > ### Comment · Reviewer_KjwZ · 2025-08-03
> > **Reply to Rebuttal**
> >
> > Thanks for the detailed reply! The reply and promised edits resolve the main weakness of the work (i.e., clarifying the roles and mechanisms of multiple attention heads) in my opinion, so I will raise the score to recommend acceptance.
> >
> > In my opinion, the one thing which may make your work better and more convincing is to actually do the synthetic experiment you suggested and show that you can precisely quantify the behavior of multiple attention sinks/tags/etc. in controlled experiments. Scientifically this may generate additional insights which could propagate to your experiments on real data. But in my opinion it's also good for quantifying and making clear in writing the intuitions you may have for real models (but which may be hard to see in messy real setups).

---

> > > ### Author Response · Authors · 2025-08-04
> > >
> > > Thank you for the follow-up, we greatly appreciate it.
> > >
> > > ---
> > >
> > > ## ... actually do the synthetic experiment you suggested and show that you can precisely quantify the behavior of multiple attention sinks/tags/etc. in controlled experiments.
> > >
> > > To directly address your suggestion, we conducted a synthetic experiment closely aligned with the setup outlined in our rebuttal:
> > >
> > > **TASK:** We modified the averaging task by introducing a second special token, [SEP2], which is inserted at a random position following the [SEP] token. The task is to compute the **sum** of two quantities:
> > >
> > > - the average of the numbers that appear after [SEP], and
> > > - the **average of the numbers that appear between [SEP] and [SEP2]**.
> > >
> > > **EMBEDDINGS:** We extended the embedding space by one dimension, such that the number tokens are embedded as $[x, -1, -1]$. The embeddings for [SEP] and [SEP2] are optimized during training.
> > >
> > > **MODEL:** We used a two-layer attention-only transformer with:
> > > - Two attention heads per layer.
> > > - Head dimension of 3.
> > > - A learned output projection $W_o$ in each layer mapping the concatenated heads back to the embedding space.
> > >
> > > Our experiments show that, at low loss, the attention heads behave as hypothesized, where in the:
> > >
> > > - **First layer**:
> > >   - One head consistently forms a single sink at the [SEP] token, isolating tokens that follow it.
> > >   - The other head forms two sinks, one at [SEP] and another at [SEP2], isolating the tokens that lie between them.
> > >
> > > - **Second layer**:
> > >   - One head approximates a uniform distribution over the number tokens after [SEP].
> > >   - The other head approximates a uniform distribution over the number tokens between [SEP] and [SEP2].
> > >
> > > The synthetic experiments show the necessity of multiple attention heads. Specifically, each attention head is dedicated to solving a specific portion of the task. In the first layer, the different heads are responsible for forming distinct sinks and tags that the model uses (constructively) in the second layer to form the two desired probability distributions.
> > >
> > > We will include visualizations of these learned attention patterns in the Appendix of the revised manuscript.
> > >
> > > To test the necessity of multiple heads, we repeated the experiment with a two-layer transformer using only one attention head per layer. Across 15 trials, none of the models converged to a low-loss solution. This provides empirical evidence that multiple attention heads are essential for solving this task.
> > >
> > > ---
> > >
> > > Again, we thank the reviewer for engaging in this discussion and remain available for further clarification until the end of the author-reviewer interaction period.

---

### Official Review · Reviewer_ZXyE · 2025-07-02

**Clarity:** 4
**Significance:** 4
**Originality:** 4
**Rating:** 5
**Confidence:** 4

**Summary:**

This paper investigates the phenomenon of "attention sinks" in Large Language Models (LLMs), where attention is disproportionately focused on a few specific tokens (like the first token or punctuation). The authors propose and validate a novel functional mechanism they term "catch, tag, release." This framework posits that attention sinks "catch" the attention of other tokens, "tag" them by adding a common directional vector to their embeddings, and then "release" these tagged embeddings back into the residual stream. This tag, as the authors demonstrate, carries semantically meaningful information that can be used by deeper layers of the model.
The paper provides extensive evidence for this mechanism through the following contributions:
1.	Visual (PCA) and quantitative ("variance explained") analyses across a wide range of model families (LLAMA, PHI, QWEN, etc.) confirm the existence and significance of this tagging behavior.
2.	Clever probing experiments on a truth-falsity dataset show that these tags can encode high-level semantic information, such as the truth value of a statement, with very high accuracy (up to 99.5%).
3.	The authors show that the mechanism is more pronounced in models fine-tuned for reasoning and is robust to architectural changes like Query-Key Normalization.
4.	A key contribution is a minimal, two-layer Transformer model designed for a sequence-averaging task. The authors prove theoretically (Theorem 7.1) that the 'catch, tag, release' mechanism is a valid solution and demonstrate that this exact mechanism emerges spontaneously through standard training.
The paper concludes that attention sinks are not an artifact or a bug, but a fundamental and functional mechanism for information propagation and labeling within LLMs.

**Questions:**

⦁	The probing experiments in Section 4 are very convincing. Are there other kinds of semantic or syntactic information encoded by these tags? For instance, in narrative text, could tags encode sentiment, character viewpoints, or tense, thereby "tagging" clauses for downstream processing?
⦁	The "Release" step is described as the tagged tokens propagating through the residual stream. Are there specific types of attention heads or MLP neurons in subsequent layers that specialize in processing tokens based on the direction of these tags?
⦁	In the theoretical model (Section 7), the "tag" and "number" components of the embedding are cleanly separated into two dimensions. In real-world models, these are obviously entangled within a high-dimensional space. The probing results suggest they are still highly separable. How "clean" is this separation in practice? Does the tag component have minimal projection onto the token's original semantic direction?
⦁	In Section 3.1, the threshold ϵ for identifying an attention sink is set to 0.2. How sensitive are the quantitative results, such as the average number of sinks (Figure 3a) and the variance explained (Figure 3b), to the choice of this hyperparameter?

**Ethical Concerns:**

["NO or VERY MINOR ethics concerns only"]

**Final Justification:**

The paper provides a functional explanation for the widely observed attention sink phenomenon. The paper is both theoretically and empirically sound. The writing is also good. I have some minor concerns but the authors have addressed them. I think the paper is ready for acceptance.

**Limitations:**

yes

**Paper Formatting Concerns:**

Nil.

**Quality:**

4

**Strengths And Weaknesses:**

Strengths
1.	The "catch, tag, release" analogy is intuitive, powerful, and provides a functional explanation for the widely observed attention sink phenomenon. It shifts the perspective from sinks being a "bug" to a "feature."
2.	The paper's claims are supported by comprehensive experiments that are both qualitatively illustrative and quantitatively robust.
⦁	The PCA visualizations in Figure 2 provide a clear and convincing visual proof of the "tagging" process, showing token embeddings clustering after the attention layer output.
⦁	The introduction of the "variance explained by tags" metric allows for a quantitative comparison across many different models, layers, and heads, firmly establishing the ubiquity of the mechanism.
⦁	The probing experiment in Section 4 is strong. By showing that the "tag" component of an activation can classify a statement's truth value far better than the non-tag or even the full activation, it provides powerful evidence that these tags carry specific, high-level semantic content.
3.	A major strength is the theoretical analysis in Section 7. By constructing a minimal problem and proving that the 'catch, tag, release' mechanism is a valid solution (Theorem 7.1), the authors provide a solid theoretical footing for their empirical observations. The demonstration that this mechanism emerges naturally through optimization (Figure 7) is a crucial result, suggesting it is a plausible and efficient strategy learned by neural networks.
4.	The paper is exceptionally well-written and organized. The introduction clearly lays out the existing theories and their shortcomings through four specific open questions (Q1-Q4). The contributions (A1-A4) directly address these questions, creating a tight and compelling narrative. The figures are clear, well-captioned, and effectively support the paper's arguments.
Weaknesses
1.	Limited Scope of Semantic Investigation: The paper brilliantly demonstrates that tags can encode truth-values. However, it leaves open the question of the full range of information these tags can carry. It would be interesting to see if this mechanism is also used for lower-level syntactic roles, coreference resolution, or other types of semantic relations.
2.	Simplicity of the Theoretical Model: As the authors commendably acknowledge in their limitations (Appendix B), the theoretical model is a highly simplified two-layer Transformer operating on 2D embeddings. While perfect for an illustrative proof-of-concept, it is a significant simplification of modern, deep, high-dimensional LLMs. The connection between the clean, emergent mechanism in this toy model and the potentially messier version in practice could be explored further.

---

> ### Author Rebuttal · Authors · 2025-07-30
>
> Thank you for your review and for providing additional ideas and suggestions as to how we can extend our experiments.
>
> ---
>
> ## Are there other kinds of semantic or syntactic information encoded by these tags? For instance, in narrative text, could tags encode sentiment…
>
> Indeed, the tagging mechanism we explore can potentially capture a broader range of semantic and syntactic information beyond just binary True/False classifications. To show this explicitly, **we extended our experiments to assess whether tags can encode sentiment**. Specifically, we construct prompts in the following format:
>
> ```
> it's a charming and often affecting journey. This statement is: POSITIVE.
> unflinchingly bleak and desperate. This statement is: NEGATIVE.
> [SENTENCE]. This statement is:"
> ```
>
> where [SENTENCE] is replaced with various examples from the SST-2 dataset. Using the same experimental setup described in Section 4, we evaluated whether tags could capture sentiment polarity. The results, presented in the table below, show that **tags indeed encode positive and negative sentiment information effectively, supporting the broader applicability of the method to a variety of different semantic tasks**.
>
> | **Probe**             | **QWEN 2.5** |    **QWEN 2.5**    |    **QWEN 2.5**    | **LLAMA-3** | **LLAMA-3.1** |
> |-----------------------|--------------|--------|--------|-------------|---------------|
> |                       | 3B           | 7B     | 14B    | 8B          | 8B            |
> | $ \theta_{\text{tag}} $       | 88.5%          | 90.0%  | 94.0%  | 87.5%       | 86.5%         |
> | $ \theta_{\text{no tag}} $     | 84.0%        | 53.0%  | 53.5%  | 81.5%       | 64.0%         |
> | $ \theta_{\text{activation}} $ | 86.5%        | 80.0%  | 90.5%    | 88.0%       | 83.0%         |
>
> ---
>
> ## Are there specific types of attention heads or MLP neurons in subsequent layers that specialize in processing tokens based on the direction of these tags?
>
> This is an excellent and non-trivial question. In principle, one could examine the cosine similarity between tag embeddings and individual neurons (i.e., rows in the MLP weight matrices of subsequent layers) to identify whether certain neurons respond selectively to tags. However, as discussed in [1], **it is unlikely that features, such as tags, are cleanly localized to individual neurons**. Instead, model features tend to be superimposed across many directions in activation space due to the model’s limited dimensionality relative to the complexity of the data it learns to encode.
>
> In this view, the directions corresponding to tags may not be easily detectable when analyzing individual neuron activations. **Instead, we hypothesize that the representation of a tag is distributed and likely spans multiple directions in the activation space, potentially spanned by the top singular vectors of the relevant weight matrices**. This suggests that tag-related information is embedded across a subspace rather than isolated to specific neurons.
>
> Probing this hypothesis would require more advanced interpretability techniques—for example, analyzing representations in the singular vector basis, using sparse autoencoders, or applying subspace probing tools from mechanistic interpretability to recover the directions associated with specific functions. We believe this is a promising direction for future research.
>
> [1] Toy Models of Superposition https://transformer-circuits.pub/2022/toy_model/index.html
>
> ---
>
> ## How "clean" is this separation in practice? Does the tag component have minimal projection onto the token's original semantic direction?
>
> We conducted additional experiments to better understand how distinct the tag component is from the token's original semantic representation. Specifically, **we measured the cosine similarity between the tag (after being mapped back into the residual stream) and the token representations of non-sink tokens to which the tag is applied, quantifiying the degree of overlap between the tag direction and the token’s inherent semantics**.
> We averaged the results over 200 prompts of 1024 tokens each, and across all layers of the model. As shown in the table below, **the cosine similarity is consistently low, especially in the LLaMA models, which suggests that the tag subspace remains largely orthogonal to the native token representations**. This indicates a relatively "clean" separation between the semantics introduced by the tag and the token's original meaning.
>
> | | Deepseek Llama 8B | Deepseek Qwen 7B | Deepseek Qwen 14B | Llama3 8B | Llama 3.1 8B | Qwen 3 8B | Qwen 3 14B | Qwen 2.5 8B | Qwen 2.5 7B | Qwen 2.5 14B |
> | :---------------- | :---------------- | :--------------- | :---------------- | :-------- | :---------- | :------- | :-------- | :-------- | :-------- | :--------- |
> | Cosine Similarity                   | -0.0674           | -0.1651          | -0.1449           | 0.0238    | 0.0099      | -0.0868  | -0.1290   | -0.1738   | -0.1581   | -0.1444    |
>
> ---
>
> ## How sensitive are the quantitative results, such as the average number of sinks (Figure 3a) and the variance explained (Figure 3b), to the choice of this hyperparameter?
>
> To assess how sensitive our results are to this design choice, **we conducted a comprehensive sensitivity analysis. Specifically, we varied the threshold across twelve values ranging from 0.03 to 0.4 and evaluated the impact on two key metrics: (1) the average number of identified sinks and (2) the average variance explained by the tags**. We will include a plot in the appendix where each point corresponds to a threshold, the x-axis represents the average number of sinks, and the y-axis reflects the variance explained.
>
> Across all models tested, we find that the **resulting curves are typically logarithmic-like: initially, decreasing the threshold adds new sinks and increases explained variance, but this gain saturates beyond a certain point**. This suggests that smaller thresholds tend to capture less meaningful or redundant tokens. The table below provides representative values from this analysis to support this observation.
>
> | $\epsilon \qquad \qquad$  | Deepseek Llama 8B (Avg. Explained Variance) | Deepseek Llama 8B (Avg. Number of Sinks) | Llama 3.1 8B (Avg. Explained Variance) | Llama 3.1 8B (Avg. Number of Sinks) |
> | :--- | :------------------------------------------ | :--------------------------------------- | :------------------------------------ | :--------------------------------- |
> | 0.03 | 0.429                                       | 3.230                                    | 0.449                                 | 5.325                              |
> | 0.04 | 0.406                                       | 2.343                                    | 0.411                                 | 3.744                              |
> | 0.05 | 0.392                                       | 1.886                                    | 0.388                                 | 2.897                              |
> | 0.06 | 0.382                                       | 1.615                                    | 0.372                                 | 2.382                              |
> | 0.07 | 0.376                                       | 1.453                                    | 0.360                                 | 2.037                              |
> | 0.08 | 0.371                                       | 1.338                                    | 0.350                                 | 1.798                              |
> | 0.09 | 0.368                                       | 1.263                                    | 0.344                                 | 1.636                              |
> | 0.1  | 0.365                                       | 1.207                                    | 0.339                                 | 1.507                              |
> | 0.15 | 0.358                                       | 1.072                                    | 0.324                                 | 1.190                              |
> | 0.2  | 0.355                                       | 1.029                                    | 0.315                                 | 1.068                              |
> | 0.3  | 0.353                                       | 0.992                                    | 0.300                                 | 0.936                              |
> | 0.4  | 0.345                                       | 0.944                                    | 0.274                                 | 0.807                              |
>
> ---
>
> We are very grateful to the reviewer for their thoughtful comments, constructive suggestions for future work, and positive evaluation of our submission. We hope that our responses and additional results further clarify the contributions and are viewed as a meaningful step forward.
> >

---

> > ### Comment · Reviewer_ZXyE · 2025-08-05
> >
> > Thanks for the response to my comments. I'll keep my positive review. Good luck!

---

### Official Review · Reviewer_RjRh · 2025-07-02

**Clarity:** 4
**Significance:** 3
**Originality:** 3
**Rating:** 5
**Confidence:** 4

**Summary:**

This paper studies attention sinks in large language models, which are tokens that receive a lot of attention, like the first token or punctuation marks. The authors argue that these sinks catch attention, add shared information to other tokens (tag), and release that information into the model’s residual stream. They provide visual and quantitative evidence for this mechanism in several popular models, show that the tags carry semantic meaning (like true/false labels), and observe that the effect is stronger in models trained for reasoning. They also introduce a small theoretical model where this behavior emerges naturally.

**Questions:**

Main Questions:
- **(Q1)** Could you examine which tokens become sinks (e.g., punctuation, logical connectors), whether they equally contribute to explained variance, and whether this changes after reasoning fine-tuning? Is it possible to create a taxonomy, possibly using bigger and different dataset? This would clarify whether the additional sinks in reasoning models are semantically meaningful.
- **(Q2)** The threshold-based sink definition is arbitrary, based on previous studies. Could you provide some intuition why this metric is indeed select all attention sinks (20% of explained variance in some cases is still low)?  Maybe you could provide some sort of sensitivity analysis over threshold and compare to alternatives (e.g., entropy or KL divergence) to ensure robustness?

Minor Questions and Comments:
- I have some doubts about value for activation probe for QWEN 2.5 3B in Table 1. How could it be that accuracy is still 50% after adding tags?
- Though one can understand it intuitively, $\theta_{activation}$ is not defined. As I believe you perform computations similar to tag/no tag, but it would be better to state it in the text

**Ethical Concerns:**

["NO or VERY MINOR ethics concerns only"]

**Final Justification:**

This is a solid, well-written study of attention sinks, and their mechanistic role. The authors provided thorough additional experiments, clear methodological justification. My concerns were addressed properly.

**Limitations:**

Yes

**Quality:**

3

**Strengths And Weaknesses:**

Strengths:
- **(S1)** Novel unified concept: the “catch, tag, release” mechanism offers a clear and intuitive interpretation of attention sink behavior. This concept unifies several previously scattered observations in the literature.
- **(S2)** Clarity and accessibility: the paper is well-written, with clear diagrams and good structure. The visualizations are intuitive and support the narrative effectively.
- **(S3)** Broad empirical scope: the authors examine the phenomenon across multiple model families (QWEN, LLAMA, Mistral, PHI), architectures (with and without QK norm), and tasks (reasoning vs. non-reasoning). This improves the generality of the claims. The experiments showing that sink-derived tags encode the truth value of statements are also a strong contribution.

Weaknesses:
- **(W1)** Lack of analysis on which tokens become attention sinks: while the paper compares the number of sinks between models (e.g., reasoning vs. pretrained), it does not analyze the types of tokens that serve as sinks. The discussion and analysis of attention sinks types could be very beneficial and provide more intuitive insights
- **(W2)** Arbitrary sink detection metric: the definition of an attention sink (Eq. 1) relies on a fixed attention threshold, but this choice is not validated. There’s no sensitivity analysis and comparison to alternative metrics (e.g., attention entropy?). This undermines the robustness of the conclusions.

---

> ### Author Rebuttal · Authors · 2025-07-30
>
> We thank the reviewer for their thoughtful review and for offering several valuable suggestions to enhance the comprehensiveness of our analysis.
>
> ---
>
> ##  Could you examine which tokens become sinks, whether they equally contribute to explained variance, and whether this changes after reasoning fine-tuning? Possible to create a taxonomy, using bigger and different dataset?
>
> We think this is a great suggestion and have **executed additional experiments that count the frequency of tokens that appear as sinks and measure the average explained variance of each token when it appears as a sink**.
> To this end, we analyzed **200 prompts of length 1024 from the QuALITY dataset**, using **both the base and reasoning-tuned versions of the Qwen 14B model**. The results, which are representative of broader trends we have observed, are now included in Section 5 and summarized in the table along with a paragraph for discussion below:
>
> | Model               | Sink String | Avg. Variance Explained | Frequency |
> |---------------------|-------------|--------------------------|-----------|
> | Qwen 2.5 14B        | BOS         | 0.28              | 328487    |
> |                     | .           | 0.25              | 5323      |
> |                     | ,           | 0.20             | 3807      |
> |                     | Ġthe        | 0.23             | 3333      |
> |                     | ĠI          | 0.23              | 3283      |
> |                     | Ġ”          | 0.24              | 3203      |
> |                     | Ġto         | 0.19              | 2603      |
> |                     | Ġbegan      | 0.20              | 2592      |
> |                     | ĠStark      | 0.18              | 2512      |
> |                     | ,”          | 0.22              | 2253      |
> | Deepseek Qwen 14B   | BOS         | 0.30             | 307600    |
> |                     | THE         | 0.31              | 50248     |
> |                     | The         | 0.30              | 47535     |
> |                     | Doctor      | 0.30              | 26928     |
> |                     | SP          | 0.30              | 24718     |
> |                     | MON         | 0.31              | 23680     |
> |                     | CAP         | 0.30              | 22832     |
> |                     | IT          | 0.29              | 22416     |
> |                     | GR          | 0.30              | 21915     |
> |                     | IMAGE       | 0.30              | 20272     |
>
> >In base models such as Qwen2.5-14B, sink formation predominantly occurs around function words (e.g., Ġthe, Ġto, ĠI) and punctuation (e.g., . or ,), likely due to their high frequency and syntactic roles. In contrast, the DeepSeek-Qwen-14B reasoning-tuned model forms attention sinks around semantically significant or task-structuring tokens, such as:
> >
> > - IMAGE suggests segmentation for multimodal inputs, grouping the token with subsequent image-related descriptions.
> > - Doctor likely marks named entities relevant for reasoning over domain-specific content.
> > - Tokens like MON, CAP, IT, and GR are plausible abbreviations or categorical labels (e.g., weekdays, captions, grades) that suggest segmentation for structured data.
> > - SP may represent speaker or section markers, important in multi-step or dialog-based reasoning.
> >
> >These sink tokens not only appear with high frequency but also explain a substantial portion of the variance, indicating their importance in structuring the model’s internal representation. The presence of such tokens—absent in the base model’s sink list—suggests that reasoning fine-tuning reorients the attention sink mechanism from syntactic attractors to semantically meaningful or task-specific units.
>
> ---
>
> ##  The threshold-based sink definition is arbitrary, based on previous studies. Could you provide some intuition why this metric is indeed select all attention sinks
>
> We agree that this is a limitation of our current methodology and that the threshold we adopt, based on prior work, does not guarantee the identification of all tokens functioning as attention sinks. To acknowledge this, **we have added the following paragraph to the Limitations section (Appendix B)**:
>
> >The threshold-based metric used to identify attention sinks does not guarantee full coverage of all such tokens. Our choice of a 0.2 threshold, while consistent with prior studies, is not necessarily optimal. As discussed in [1], there is currently no principled method for determining this threshold.
>
> To assess how sensitive our results are to this design choice, we conducted a comprehensive sensitivity analysis. Specifically, **we varied the threshold across twelve values ranging from 0.03 to 0.4 and evaluated the impact on two key metrics: (1) the average number of identified sinks and (2) the average variance explained by the tags.** We will include a plot in the Appendix where each point corresponds to a threshold, the x-axis represents the average number of sinks, and the y-axis reflects the variance explained.
>
> Across all models tested, **we find that the resulting curves are typically logarithmic-like: initially, decreasing the threshold adds new sinks and increases explained variance, but this gain saturates beyond a certain point**. This suggests that smaller thresholds tend to capture less meaningful or redundant tokens. The table below provides representative values from this analysis to support this observation.
>
>
> | $\epsilon \qquad \qquad$  | Deepseek Llama 8B (Avg. Explained Variance) | Deepseek Llama 8B (Avg. Number of Sinks) | Llama 3.1 8B (Avg. Explained Variance) | Llama 3.1 8B (Avg. Number of Sinks) |
> | :--- | :------------------------------------------ | :--------------------------------------- | :------------------------------------ | :--------------------------------- |
> | 0.03 | 0.429 | 3.230 | 0.449 | 5.325 |
> | 0.04 | 0.406 | 2.343 | 0.411 | 3.744 |
> | 0.05 | 0.392 | 1.886 | 0.388 | 2.897 |
> | 0.06 | 0.382 | 1.615 | 0.372 | 2.382 |
> | 0.07 | 0.376 | 1.453                                    | 0.360                                 | 2.037                              |
> | 0.08 | 0.371                                       | 1.338                                    | 0.350                                 | 1.798                              |
> | 0.09 | 0.368                                       | 1.263                                    | 0.344                                 | 1.636                              |
> | 0.1  | 0.365                                       | 1.207                                    | 0.339                                 | 1.507                              |
> | 0.15 | 0.358                                       | 1.072                                    | 0.324                                 | 1.190                              |
> | 0.2  | 0.355                                       | 1.029                                    | 0.315                                 | 1.068                              |
> | 0.3  | 0.353                                       | 0.992                                    | 0.300                                 | 0.936                              |
> | 0.4  | 0.345                                       | 0.944                                    | 0.274                                 | 0.807                              |
>
> ---
>
> ## … compare to alternatives (e.g., entropy or KL divergence)
>
> We initially considered using alternative metrics such as entropy or KL divergence to identify attention sinks. However, attention sinks are defined over columns of the attention probability matrix, that is, a sink corresponds to a token that consistently receives high attention across multiple rows. **Since the columns are not normalized, the sink column does not represent a valid probability distribution. As such, entropy or KL divergence cannot be meaningfully computed over columns.**
>
> Likewise, using row-wise entropy to detect focused attention would be unable to capture attention sinks. **Low entropy across rows merely indicates that each token is attending strongly to a few others, but it says nothing about which tokens are being attended to.** For example, if the attention matrix is close to diagonal, each row has low entropy, yet no token consistently receives attention—i.e., no sink exists.  For this reason, we opted for a threshold-based metric that directly evaluates whether certain tokens receive consistently high aggregate attention, better aligning with the conceptual definition of attention sinks.
>
> ---
>
> ## How could it be that accuracy is still 50% after adding tags?
>
> We have double-checked the data, and the result is indeed accurate for that specific attention head. One possible explanation is that the tag portion of the activation has a significantly smaller norm compared to the non-tag portion. As a result, the classifier may effectively ignore the tag component due to its negligible contribution, leading to no improvement in accuracy despite the presence of tags.
>
> ---
>
> ## $\theta_{activation}$  is not defined, would be better to state it in the text
>
> Thank you for pointing out a potential point of unclarity. We have included in the Appendix the full equations and descriptions of the probe.
>
> ---
>
> We thank the reviewer for suggesting several valuable directions for additional measurements which have enhanced the comprehensiveness of our analysis. If the reviewer finds that these additions strengthen the paper, we would be grateful for any reconsideration of the score.
>
> >
>
> [1] Gu et al. (2024). When Attention Sink Emerges in Language Models: An Empirical View.

---

> > ### Comment · Reviewer_RjRh · 2025-08-03
> >
> > Thank you for the detailed and thoughtful response. Most of my concerns have been fully addressed, and I am ready to raise my score.
> >
> > I do have one clarifying question: in your discussion of the 50% accuracy result, you mention that this applies to a specific attention head. Could you clarify how this head was selected? As I understand, you find head where tags performance is the best. But can we say that if head's full activation/non-tag activations classify True/False, then only tag should also perform such classification correctly?

---

> > > ### Author Response · Authors · 2025-08-04
> > >
> > > Thank you for the follow-up and for reconsidering your score. We greatly appreciate it.
> > >
> > > ---
> > >
> > > ## Could you clarify how this head was selected?
> > >
> > > We selected attention heads based on two complementary criteria:
> > >
> > > 1. **High classification performance from tag-only probes $\theta_{\text{tag}}$**, and
> > > 2. **Low performance from non-tag probes $\theta_{\text{no tag}}$**.
> > >
> > > These criteria directly support the goal of Section 4. Specifically, (1) demonstrates that the model is leveraging the 'catch, tag, release' mechanism to transmit semantically meaningful information, while (2) provides evidence that the 'catch, tag, release' mechanism is not merely redundantly disseminating information that is also encoded in the representations of non-sink tokens.
> > >
> > > ---
> > >
> > > ## Can we say that if a head’s full activation or non-tag component can classify True/False, then the tag should also be able to do so?
> > >
> > > Not necessarily. The 'catch, tag, release' mechanism transmits various types of information, not just the truth value of a sentence. While some heads disseminate True/False information via their tags, others may propagate different semantic information, such as sentiment. In such cases, full or non-tag activations may perform well on truth classification, but tag-based probes will not perform well on that task -- they will instead perform well on sentiment classification.
> > >
> > > ---
> > >
> > > We hope this clarifies the reviewer's question. Please let us know if any further elaboration would be helpful and we remain available for discussion.

---

> > > > ### Comment · Reviewer_RjRh · 2025-08-05
> > > >
> > > > Thank you for the clarification, this addresses my concern. I now have no remaining issues, and I am raising my score to 5.

---

### Official Review · Reviewer_Z8vb · 2025-07-03

**Clarity:** 3
**Significance:** 3
**Originality:** 3
**Rating:** 4
**Confidence:** 3

**Summary:**

This paper investigates the phenomenon of "attention sinks" in large language models, proposing that these sinks implement a "catch, tag, release" mechanism. In this process, specific tokens act as sinks to "catch" the attention of other tokens, "tag" them by adding their value vectors to the embeddings, and "release" this new information into the model's residual stream for use in subsequent layers.
Through probing experiments, they demonstrate that the tags carry semantically meaningful information, such as the truth-value of a statement. The study finds that this mechanism is more prominent in models fine-tuned for reasoning than in their pretrained counterparts. It also shows that attention sinks persist even in models with query-key (QK) normalization, an architecture that normalizes the query and key activations and may potentially suppress attention sinks. To ground these empirical findings, the paper introduces a minimal theoretical problem and provides a constructive proof showing that a two-layer transformer can solve it by explicitly implementing the "catch, tag, release" mechanism.

**Questions:**

- The paper states in Section 7.1 that the theoretical model uses causal attention. However, the proof in Appendix C.1 appears to contradict this. Specifically, in line 580, the equation for the attention weight $A_{i,t}^1$. (for query $i$ attending to key $t$) defines the denominator as a sum over all keys from $k = 1$ to $T$. For causal attention, this sum should only be over keys from $k = 1$ to $i$.

- I notice there is a discrepancy between theoretical constructions and empirical training results. The theorem and its proof rely on the assumption that the tag value $s_{tag} \to \infty$. However, in my attempt to reproduce the results using the provided code, I observed that the learnable parameter $s_{tag}$, initialized at $10$ as per Appendix D, does not grow large and instead decreases slightly to around $9.3$ during training. Furthermore, I was unable to reproduce the clear averaging behavior shown in Figure 7(c).

- Minor: there is a typo in line 622 as the initialization value should be $10$ instead of $-10$ according to the codes.

**Ethical Concerns:**

["NO or VERY MINOR ethics concerns only"]

**Final Justification:**

Most of my concerns have been resolved so I'm happy to raise the score to 4.

**Limitations:**

Yes, the authors address some limitations in Appendix B regarding the theoretical results in Section 7, but the discussion could be more comprehensive by also including the limitations of the empirical analyses in Sections 5 and 6.

**Quality:**

3

**Strengths And Weaknesses:**

Strenghts
- The paper introduces a novel "catch, tag, release" mechanism to explain the function of attention sinks in LLMs. This work contributes to a deeper understanding of LLMs' behavior, enhancing the interpretability.

- The proposed mechanism is supported by a comprehensive empirical investigation. The authors validate the "catch, tag, release" mechanism across various LLM families and sizes (including Qwen, Phi, and Llama), demonstrating the generality of their findings. The claims are complemented by probing experiments that convincingly show that the tags generated by attention sinks encode semantically meaningful information.

Weaknesses
- While the core "catch, tag, release" mechanism is well-supported, the analysis for two of the claims (A3 and A4) seems preliminary and less developed.
   - Claim A3 (Section 5): The paper shows that attention sinks and the proposed mechanism are more prevalent in reasoning models than in their pretrained counterparts. However, the authors do not go further to explain how this increased prevalence contributes to the models' reasoning abilities. Without this deeper analysis, the claim remains an observation.

  - Claim A4 (Section 6): The analysis of models with QK normalization is limited to showing that the number of attention sinks does not decrease without further discussion. Moreover, the paper does not verify whether the "catch, tag, release" mechanism is still present in these models. This omission makes the conclusion in this section less impactful.

Overall, while the core contributions of the paper are strong, the analysis in Section 5 and 6 is insufficient. These sections would need more evidence and discussion to be considered on par with the paper's primary findings.

---

> ### Author Rebuttal · Authors · 2025-07-30
>
> We would first like to thank the reviewer for their thorough and thoughtful analysis of the manuscript, its appendices, and the supplementary material, which we have leveraged to improve the overall quality of our work.
>
> ---
>
> ## The analysis of models with QK normalization is limited, ... the paper does not verify whether the "catch, tag, release" mechanism is still present in these models.
>
> We agree with the reviewer that the omission of measurements relating to models with QK norm was an oversight in our original manuscript. To provide evidence that the 'catch, tag, release' mechanism is still present in models with QK norm, **we have now added measurements for Qwen 3 8B, and 14B showing their sink count** (using violin plots), **variance explained by tags** (using box and whiskers plots), as presented in Section 3.2 Figure 3. We summarize these results in the table below, which averages across all attention heads.
>
> | **Probe**                |  **QWEN 3**       | **QWEN 3**  |
> |--------------------|--------------|-------|
> |                    | **8B**| **14B**|
> | Avg. Number of Sinks        |  1.03 | 1.04 |
> | Avg. Variance Explained     | 0.26 | 0.25 |
>
> Additionally, we **also executed the probing experiments provided in Section 4 on the Qwen 3 models** which are presented in the table below:
>
> | **Probe**                |  **QWEN 3**       | **QWEN 3**  |
> |--------------------|--------------|-------|
> |                    | **8B**| **14B**|
> | $\theta_{\text{tag}}$        | 100% | 87.0% |
> | $\theta_{\text{no tag}}$     | 50.5% | 64.5% |
> | $\theta_{\text{activation}}$ | 56.0% | 82.0%   |
>
> To summarize, the above results follow the same trends as the models without QK norm, suggesting that the 'catch, tag, release’ mechanism is still present in these models.
>
> ---
>
> ## The authors do not go further to explain how this increased prevalence contributes to the models' reasoning abilities.
>
> We thank the reviewer for highlighting the need for a deeper analysis supporting Claim A3. In response, we have **added a new experiment to Section 5 that includes the most frequent attention sink tokens in both reasoning-tuned and base (non-reasoning) versions of the Qwen 2.5 14B model**:
>
> | Model               | Sink String | Avg. Variance Explained | Frequency |
> |---------------------|-------------|--------------------------|-----------|
> | Qwen 2.5 14B        | BOS         | 0.28              | 328487    |
> |                     | .           | 0.25              | 5323      |
> |                     | ,           | 0.20             | 3807      |
> |                     | Ġthe        | 0.23             | 3333      |
> |                     | ĠI          | 0.23              | 3283      |
> |                     | Ġ”          | 0.24              | 3203      |
> |                     | Ġto         | 0.19              | 2603      |
> |                     | Ġbegan      | 0.20              | 2592      |
> |                     | ĠStark      | 0.18              | 2512      |
> |                     | ,”          | 0.22              | 2253      |
> | Deepseek Qwen 14B   | BOS         | 0.30             | 307600    |
> |                     | THE         | 0.31              | 50248     |
> |                     | The         | 0.30              | 47535     |
> |                     | Doctor      | 0.30              | 26928     |
> |                     | SP          | 0.30              | 24718     |
> |                     | MON         | 0.31              | 23680     |
> |                     | CAP         | 0.30              | 22832     |
> |                     | IT          | 0.29              | 22416     |
> |                     | GR          | 0.30              | 21915     |
> |                     | IMAGE       | 0.30              | 20272     |
>
> This table reveals not only a quantitative increase in sink prevalence, but also a qualitative shift in which tokens form sinks, indicating a meaningful structural adaptation in how the 'catch, tag, release' mechanism operates under reasoning fine-tuning as **we describe in the additional paragraph that will be added to the manuscript**:
>
> >In base models such as Qwen 2.5 14B, sink formation predominantly occurs around function words (e.g., Ġthe, Ġto, ĠI) and punctuation (such as . or ,), likely due to their high frequency and syntactic roles. In contrast, the DeepSeek Qwen 14B reasoning-tuned model forms attention sinks around semantically significant or task-structuring tokens, such as:
> > - IMAGE suggests segmentation for multimodal inputs, grouping the token with subsequent image-related descriptions.
> > - Doctor likely marks named entities relevant for reasoning over domain-specific content.
> > - Tokens like MON, CAP, IT, and GR are plausible abbreviations or categorical labels (e.g. weekdays, captions, grades) that suggest segmentation for structured data.
> > - SP may represent speaker or section markers, important in multi-step or dialog-based reasoning.
> >
> > These sink tokens not only appear with high frequency but also explain a substantial portion of the variance, indicating their importance in structuring the model’s internal representation. **The presence of such tokens, absent in the base model’s sink list, suggests that reasoning fine-tuning reorients the attention sink mechanism from syntactic attractors to semantically meaningful or task-specific units.**
>
> This new evidence strengthens Claim A3 by showing that attention sinks in reasoning-tuned models are not just more frequent, they are repurposed to segment, tag, and organize input in ways that directly support reasoning behaviors such as modality transitions, entity tracking, and logical clause separation. The 'catch, tag, release' mechanism thus adapts in both frequency and function to facilitate reasoning.
>
> We also now **explicitly acknowledge in the Limitations section that this connection remains observational**:
>
> > While our findings in Section 5 strongly suggest a structural role for sinks in reasoning, they do not yet establish a causal link between the two. This remains an important direction for future work.
>
> ---
>
> ## In line 580, the equation for the attention weight... defines the denominator as a sum over all keys from $k=1$ to $T$ .
>
> Thank you for catching this error. **The summation should indeed be from $k=1$ to $ k=i$ instead of from $k=1$ to $k=T$** (which would change the first two lines of that derivation). However, **this does not impact the later two equations in that derivation so the ultimate limit and result is still correct**. We have also carefully gone over all other equations in the proof to ensure that this typo does not occur elsewhere.
>
> ---
>
> ## Unable to reproduce the clear averaging behaviour shown in Figure 7(c)
>
> If no clear averaging behavior emerges, **this indicates that optimization failed to converge to a low-loss solution**. We have observed this same issue in our own experiments. However, in all successful runs that do converge, we consistently observe the emergence of the 'catch, tag, and release’ mechanism and the clear averaging shown in Figure 7(c). **To clarify these issues, we have added the following paragraph to the limitations section in Appendix B**:
>
> > Although the theoretical model is simple, the optimization process does not always reach a low-loss solution. We found that convergence depends on how the $s_{\text{tag}}$ parameter is initialized, with larger initial values generally leading to more consistent success. The table below shows how often the model achieved a high fit ($R^2 > 0.95$) across 10 runs for different starting values of $s_{\text{tag}}$:
>
> | $s_{\text{tag}}$ initialization | Success Rate ($R^2>0.95$) |
> |----------------------|----------------------|
> | 10                    | 4/10                |
> | 6                   |     3/10           |
> | 1                    |     0/10            |
>
> ---
>
> ## The learnable parameter, initialized at 10 as per Appendix D, does not grow large and instead decreases
>
> The theoretical result provides sufficient conditions for the emergence of the mechanism, but these are not necessary. Empirically, **we observe that the mechanism can still emerge under different dynamics**. For instance, instead of $s_{\text{tag}}$ diverging while the weights remain fixed, it is also possible for $s_{\text{tag}}$ to remain fixed while the weights shrink. Fully characterizing all such cases is non-trivial. However, **the key point is that the 'catch, tag, release' mechanism consistently emerges in this toy setting**. Moreover, **$s_{\text{tag}}$ is initialized to a value already $10\times$ larger than the second coordinate of the number tokens, which may reduce the need for it to grow further**.
>
> ---
>
> ## There is a typo in line 622
>
> Thank you for bringing this to our attention, **we have since revised the appendix to correct this typo from -10 to 10**.
>
> ---
>
> We extend our gratitude to the reviewer for their careful analysis of our paper and for highlighting issues that we have overlooked. Should the reviewer think that we have taken sufficient action to address their concerns, we would be most appreciative of any reconsideration of their score.
> >

---

> > ### Comment · Reviewer_Z8vb · 2025-08-05
> >
> > Thank you for the response. Most of my concerns have been addressed, and I am happy to raise my score. Moreover, I have two additional questions:
> >
> > - What is your input data when testing attention sinks in the reasoning model? The sink strings are hard to interpret without knowing the data.
> >
> > - In your theoretical model, why does it sometimes fail to converge? Is this related to optimization hyperparameters like the learning rate, or are there other reasons?

---

> > > ### Author Response · Authors · 2025-08-06
> > >
> > > Thank you for responding to our rebuttal, we greatly appreciate it!
> > >
> > > ---
> > >
> > > ## What is your input data when testing attention sinks in the reasoning model? The sink strings are hard to interpret without knowing the data.
> > >
> > > For this experiment, we utilized a set of 200 prompts, sourced from the QuALITY dataset [1], which we truncate to 1024 tokens. We specfically chose this dataset as it is used as a benchmark for long-context multiple-choice reading comprehension, consisting of questions based on full-length documents such as stories and articles.
> > >
> > > [1] Pang et al. (2021). QuALITY: Question Answering with Long Input Texts, Yes!.
> > >
> > > ---
> > >
> > > ## In your theoretical model, why does it sometimes fail to converge? Is this related to optimization hyperparameters like the learning rate, or are there other reasons?
> > >
> > >  In our synthetic experiments, we observed that convergence is quite sensitive to the initialization of model parameters. To isolate this effect, we kept all other factors constant, including learning rate, weight decay, and training samples, and varied only the parameter initialization. Even when fixing the embedding of the [SEP] token (which we already set to aid in convergence), the initialization of the remaining parameters still played a crucial role in determining whether the model converged to a low-loss solution.
> > >
> > >  ---
> > >
> > >  We hope this has helped answer the reviewer's questions and remain available until the end of the author-reviwer discussion period. Thank you!

---

### Author Response · Authors · 2025-08-09

As the discussion period draws to a close, we would like to thank all four reviewers for their suggestions, comments, and observations, which have made this a very productive review process and led to meaningful improvements in our work. We especially appreciate that each reviewer actively engaged with us during the discussion period beyond just acknowledging the rebuttals. Thank you!

---

### Decision · Program_Chairs · 2025-09-17

**Decision:**

Accept (poster)

**Comment:**

The paper argues that attention sinks implement a three-step mechanism: catch, tag, release. Sink tokens attract attention, copy their value vectors as tags into many tokens, and those tags later guide computation by the residual stream. The claim is supported by a variance-explained metric, visualizations, semantic probes, cross‑family coverage, and a minimal constructive theory. Reviewers asked for deeper rigor around sink detection thresholds, multi‑head interactions, and the link to reasoning. The authors responded with additional analyses (threshold sensitivity, sink‑token taxonomy, QK-norm probes on Qwen-3), sentiment probes, orthogonality across heads, and a new multi‑head synthetic experiment. The reviews converged to an overall positive.